# Genome-wide screen reveals Rab12 GTPase as a critical activator of Parkinson's disease-linked LRRK2 kinase

**Herschel S Dhekne[1,2†], Francesca Tonelli[2,3†], Wondwossen M Yeshaw[1,2†], Claire Y Chiang[1,2†], Charles Limouse[1], Ebsy Jaimon[1,2], Elena Purlyte[3‡], Dario R Alessi[2,3], Suzanne R Pfeffer[1,2\***

[1]Department of Biochemistry, Stanford University School of Medicine, Stanford, United States; [2]Aligning Science Across Parkinson's (ASAP) Collaborative Research Network, Stanford, United States; [3]MRC Protein Phosphorylation and Ubiquitylation Unit, University of Dundee, Dundee, United Kingdom

**Abstract** Activating mutations in the leucine-rich repeat kinase 2 (LRRK2) cause Parkinson's disease. LRRK2 phosphorylates a subset of Rab GTPases, particularly Rab10 and Rab8A, and we showed previously that these phosphoRabs play an important role in LRRK2 membrane recruitment and activation (Vides et al., 2022). To learn more about LRRK2 pathway regulation, we carried out an unbiased, CRISPR-based genome-wide screen to identify modifiers of cellular phosphoRab10 levels. A flow cytometry assay was developed to detect changes in phosphoRab10 levels in pools of mouse NIH-3T3 cells harboring unique CRISPR guide sequences. Multiple negative and positive regulators were identified; surprisingly, knockout of the *Rab12* gene was especially effective in decreasing phosphoRab10 levels in multiple cell types and knockout mouse tissues. Rab-driven increases in phosphoRab10 were specific for Rab12, LRRK2-dependent and PPM1H phosphatase-reversible, and did not require Rab12 phosphorylation; they were seen with wild type and pathogenic G2019S and R1441C LRRK2. As expected for a protein that regulates LRRK2 activity, Rab12 also influenced primary cilia formation. AlphaFold modeling revealed a novel Rab12 binding site in the LRRK2 Armadillo domain, and we show that residues predicted to be essential for Rab12 interaction at this site influence phosphoRab10 and phosphoRab12 levels in a manner distinct from Rab29 activation of LRRK2. Our data show that Rab12 binding to a new site in the LRRK2 Armadillo domain activates LRRK2 kinase for Rab phosphorylation and could serve as a new therapeutic target for a novel class of LRRK2 inhibitors that do not target the kinase domain.

**\*For correspondence:**
pfeffer@stanford.edu

†These authors contributed equally to this work

**Present address:** ‡University of Texas Southwestern Medical Center, Dallas, United States

**Competing interest:** The authors declare that no competing interests exist.

## Editor's evaluation

LRRK2 is a multi-domain kinase and is known to phosphorylate a subset of Rab proteins involved in intracellular trafficking, and Parkinson's disease-linked mutations increase this phosphorylation. How LRRK2 becomes activated is a major question in the field. This highly interesting work adds a new layer to our mechanistic understanding of this complex protein, revealing that binding of Rab12 to LRRK2 stimulates its ability to phosphorylate Rab10, a conclusion that is supported by extensive and robust evidence from a wide array of approaches.

## Introduction

Activating mutations in the large, multidomain, leucine-rich repeat kinase 2 (LRRK2) cause inherited Parkinson's disease and lead to the phosphorylation of a subset of Rab GTPases (*Alessi and Sammler,*

*2018*; *Vides et al., 2022*; *Pfeffer, 2023*), particularly Rab8A and Rab10 (*Steger et al., 2016*; *Steger et al., 2017*). Rab GTPases function in all steps of membrane trafficking by binding to specific effector proteins in their GTP-bound states (*Pfeffer, 2017*); they are well known for linking motor proteins to transport vesicles and facilitating the transport vesicle docking process.

LRRK2 phosphorylates a single threonine or serine residue in substrate Rab GTPase switch II domains, and this modification blocks the ability of Rabs to be activated by their cognate guanine nucleotide exchange factors, recycled by GDI protein, or bind to their effector proteins (*Steger et al., 2016*; *Steger et al., 2017*). Instead, phosphorylated Rabs bind to a new set of phosphoRab effectors that include RILPL1, RILPL2, JIP3, JIP4, and MyoVa proteins (*Steger et al., 2017*; *Waschbüsch et al., 2020*; *Dhekne et al., 2021*). Although only a small percentage of a given Rab protein is LRRK2 phosphorylated at steady state (*Ito et al., 2016*), binding to phosphoRab effectors has a dominant and powerful effect on cell physiology and can interfere with organelle motility in axons (*Boecker et al., 2021*), primary ciliogenesis (*Dhekne et al., 2018*; *Sobu et al., 2021*; *Khan et al., 2021*), and centriolar cohesion (*Lara Ordóñez et al., 2021*).

We have identified a feed-forward pathway that recruits LRRK2 to membranes and can hold it there to enhance subsequent Rab GTPase phosphorylation (*Vides et al., 2022*). As described in greater detail below, the large multidomain LRRK2 kinase relies on its N-terminal Armadillo domain to associate with membranes. The Armadillo domain contains two substrate Rab binding sites that recruit and anchor LRRK2 on membranes: one for non-phosphorylated Rab proteins and another that can bind LRRK2-phosphorylated Rab8A and Rab10. The presence of two binding sites increases the avidity of LRRK2 for membranes and holds the kinase on membrane surfaces to facilitate subsequent Rab phosphorylation (*Vides et al., 2022*).

We present here an unbiased, genome-wide CRISPR screen in mouse NIH-3T3 cells undertaken to identify regulators of the LRRK2 pathway. Of the multiple positive and negative hits identified, Rab12 was the most potent regulator of LRRK2 activity when either depleted from cells or overexpressed. We show further our surprising discovery of a third LRRK2 Rab12 binding site in the Armadillo domain that includes residues E240 and S244; site #3 mutations predicted to block Rab12 binding fail to bind Rab12 and show decreased phosphoRab10 levels, consistent with a critical role for Rab12 in LRRK2 activation.

## Results

The pooled CRISPR screen to identify modulators of LRRK2 activity utilized mouse NIH-3T3 cells in conjunction with the pooled Brie guide RNA (gRNA) mouse library consisting of 78,637 gRNAs targeting 19,674 genes and an extra 1000 control gRNAs. (A highly detailed protocol can be found on protocols.io; *Dhekne et al., 2022a*). Briefly, a pooled 'library' of Cas9-expressing cells is first generated, each cell harboring a different gene knockout. Genes encoding negative regulators of the LRRK2- phosphoRab10 pathway will increase phosphoRab10 staining when knocked out, and genes encoding positive regulators will decrease phosphoRab10 when knocked out. Fixed cells are stained with an antibody that specifically and sensitively detects phospho-Thr73-Rab10 (hereafter referred to as phosphoRab10) and then sorted by flow cytometry to separate cells based on phosphoRab10 content. Gene knockouts responsible for changes in phosphoRab10 levels are then identified by genomic sequencing of cells with higher or lower than normal phosphoRab10 levels.

*Figure 1A* shows an example of flow cytometry of anti-phosphoRab10 stained, control mouse NIH-3T3 cells analyzed under baseline conditions (blue) in relation to MLi-2-treated, LRRK2-inhibited cells (green), secondary antibody-only-stained cells (black dashed line), or LRRK2-hyperactivated, nigericin-treated NIH-3T3 cells (pink; *Kalogeropulou et al., 2020*). The flow cytometry resolution of cells with differing phosphoRab10 levels enabled us to collect the highest 7.5% phosphoRab10 signal and lowest 5% signal and compare these enriched cell populations with unsorted cells. Critical to the success of this method is the ability to obtain non-clumped cells after antibody fixation; otherwise, the average fluorescence of clumps will obscure true hits.

Statistical analysis of sequencing data from the cells with the lowest phosphoRab10 signal confirmed the success of the screen in that loss of *Lrrk2*, *Rab10*, and the *Rabif* Rab10 chaperone gene (*Gulbranson et al., 2017*) had the most significant impact on phosphoRab10 expression, as would be expected (*Figure 1B* and *Figure 1—figure supplement 1*). Similarly, loss of the *Chm* gene that is needed for Rab prenylation also led to decreased phosphoRab10. Independent revalidation of

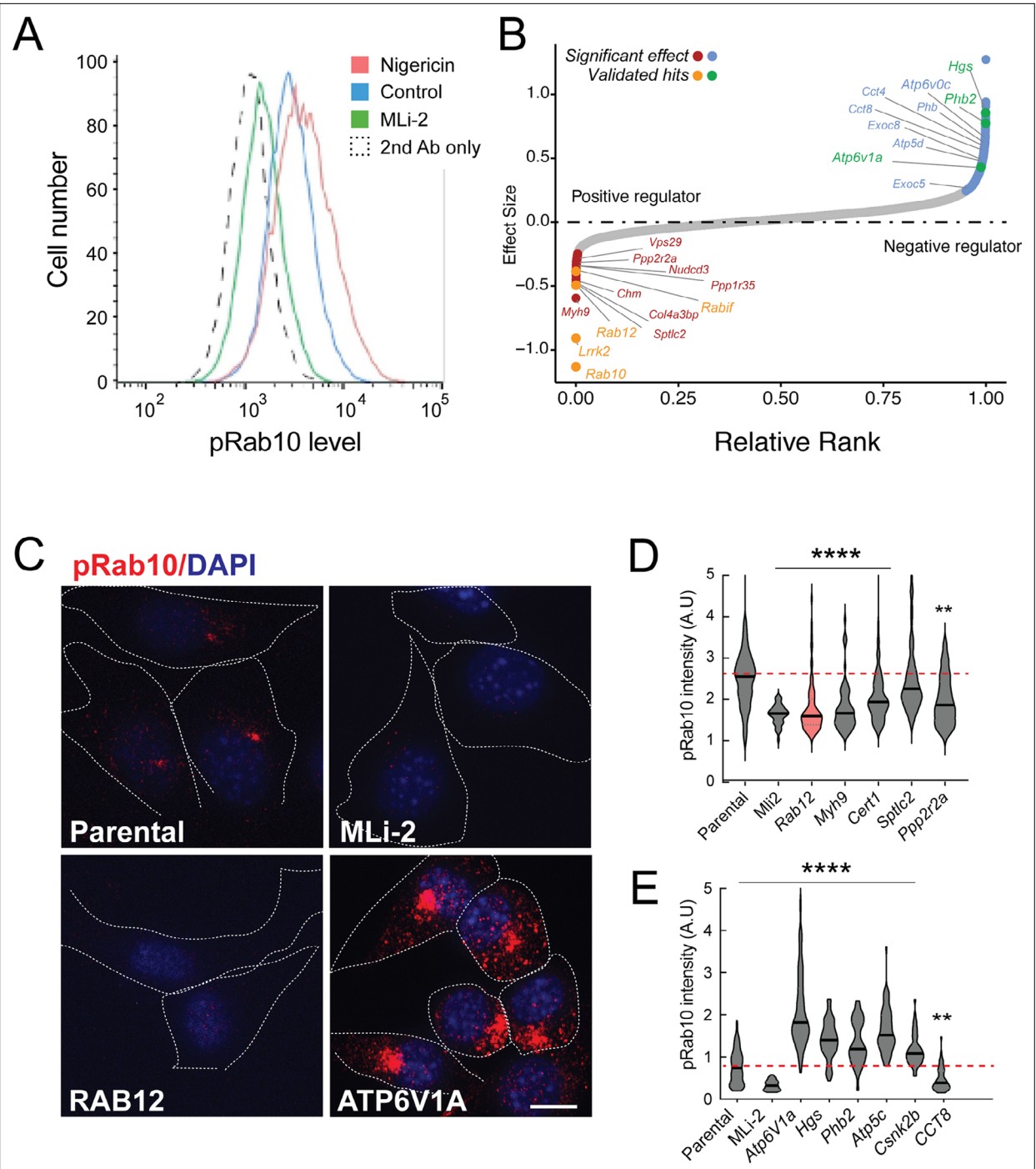

**Figure 1.** A flow cytometry-based, genome-wide CRISPR screen in NIH-3T3-Cas9 cells to reveal modifiers of the LRRK2-phosphoRab10 pathway. (**A**) Phosphorylated Rab10 was detected by flow cytometry after staining cells using anti-phosphoRab10 antibody, either at steady state (control, blue) or in the presence of 4 μM nigericin for 3 hr (red) or 200 nM MLi-2 for 2 hr (green). 10,000 cells were analyzed under each of the indicated conditions. (**B**) Statistical analysis of the genome-wide screen. After infection with a lentiviral genome-wide CRISPR-Cas9 sgRNA library, genes when knocked out that reduced (left) or increased (right) phosphoRab10 intensity are indicated on the volcano plot where the X-axis is $\log_2$-fold change and Y-axis shows the false discovery rate (FDR)-corrected confidence scores. Genes highlighted are the top positive and negative regulators. (**C, D**) Validation of hits in NIH-3T3-Cas9 cells by immunofluorescence microscopy. (**C**) PhosphoRab10 was detected by immunofluorescence microscopy in early passage NIH-3T3-Cas9 cells that express lentivirus transduced sgRNAs against the indicated gene after 3 d of puromycin selection. Scale bar = 10 μm. (**D, E**) Quantitation of phosphoRab10 fluorescence in cells in which the indicated genes are knocked out. p-values: ****<0.0001; **0.0088; n > 100 cells counted in two independent experiments.

*Figure 1 continued on next page*

*Figure 1 continued*

The online version of this article includes the following figure supplement(s) for figure 1:

**Figure supplement 1.** Guide RNA enrichment for CRISPR screen.

**Figure supplement 2.** Validation of hits in NIH-3T3-Cas9 cells by microscopy.

the most significant top hits in NIH-3T3 cells (*Figure 1C–E* and *Figure 1—figure supplement 2*) by creating individually knocked out cell lines confirmed most of them, and as will be described below, revealed an unexpected role for Rab12 GTPase.

In addition to Rab12, knockout of genes, including *Myh9, Cert1, Sptlc2, Ppp2r2a, Ppp1r35,* and *Nudcd3,* also decreased phosphoRab10 intensity by immunofluorescence microscopy, suggesting that the corresponding gene products are also positive regulators of LRRK2 function (*Figure 1B–D* and *Figure 1—figure supplement 1*). ER-localized SPTLC2 (serine palmitoyl transferase) is the rate-limiting enzyme in ceramide synthesis, and CERT1 is critical for ceramide transfer from the ER to the Golgi complex. How ceramide synthesis and transport relate to LRRK2 activity will be addressed in future work; chemical inhibition of SPTLC2 with myriocin did not yield a similar phenotype, suggesting that the role of this pathway in phosphoRab10 regulation may be more complex. PPP2R2A was shown previously to similarly influence phosphoRab10 levels in a phosphatome-wide screen to identify phosphoRab10 phosphatases (*Berndsen et al., 2019*). PPP1R35 was not tested in that screen, but like MYH9, it is involved in primary cilia assembly, and their pericentriolar localizations suggest a connection with phosphoRab10 biology. NUDCD3 stabilizes the dynein intermediate chain and is likely important for concentrating phosphoRab10 at the mother centriole (*Zhou et al., 2006*; *Cai et al., 2009*). Finally, 14-3-3 proteins such as YWHAE are known to bind LRRK2 via pSer910 and pSer935 (*Nichols et al., 2010*) and may stabilize LRRK2 protein.

Knockout of several genes hyperactivated LRRK2 activity and phosphoRab10 levels: these include *Atp6v1A, Atp6v0c, Hgs, Phb2, Atp5c,* and *Csnk2b* (*Figure 1B, C and E* and *Figure 1—figure supplement 1*). The ATP6 proteins are non-catalytic subunits of the vacuolar ATPase needed for lysosome acidification; their deletion presumably has similar effects as bafilomycin that greatly increases LRRK2 activity (*Wang et al., 2021*). HGS is also known as HRS and is part of the ESCRT-0 complex; loss of HRS function interferes with autophagic clearance and causes ER stress (*Oshima et al., 2016*). PHB1/2 are inner mitochondrial membrane mitophagy receptors that are required for Parkin-induced mitophagy in mammalian cells (*Wei et al., 2017*). Work from Ganley and colleagues has shown an inverse correlation between LRRK2 activity and mitochondrial turnover (*Singh et al., 2021*). ATP5C1 is part of the mitochondrial ATP synthase complex V; casein kinase 1 alpha has been shown to phosphorylate LRRK2 (*Chia et al., 2014*) but a role for casein kinase 2B is not yet clear. As reported previously by many other groups, lysosomal and mitochondrial stress increased phosphoRab10 levels.

## Loss of Rab12 impacts phosphoRab10 generation

*Figure 2A* compares the levels of endogenous phosphoRab10 and total Rab10 in parental NIH-3T3 cells, parental cells treated with MLi-2 LRRK2 inhibitor, and a pooled NIH-3T3 cell line in which Rab12 has been knocked out. Quantitation of these data confirmed a roughly fivefold decrease in phosphoRab10 levels under these conditions (*Figure 2B*). This was entirely unexpected as prior studies on Rab29, a protein that can activate apparent LRRK2 activity under conditions of protein overexpression (*Liu et al., 2018*; *Purlyte et al., 2018*); loss of Rab29 did not alter phosphoRab10 levels in a Rab29 mouse knockout model in any tissue analyzed or derived mouse embryonic fibroblasts (MEFs) (*Kalogeropulou et al., 2020*). We did not analyze Rab8A phosphorylation as the available antibody detects multiple phosphorylated Rab proteins (*Steger et al., 2017*).

To confirm these data in an animal model, we analyzed cells and tissues derived from *Rab12* knockout mice generated by the Knockout Mouse Phenotyping Program at The Jackson Laboratory using CRISPR technology (*Figure 2—figure supplements 1 and 2*). Immunoblotting analysis of MEFs confirmed that the heterozygous and homozygous knockouts expressed the expected 50 or 100% loss of Rab12 protein (*Figure 2C*). MEFs derived from homozygous knockout animals showed as much as 50% decrease in phosphoRab10 levels as detected by immunoblots from multiple clones (*Figure 2D*); specificity of the detection method was confirmed upon addition of the MLi-2 LRRK2 inhibitor that abolished all phosphoRab10 signals. PhosphoRab7, the product of LRRK1 action (*Hanafusa et al.,*

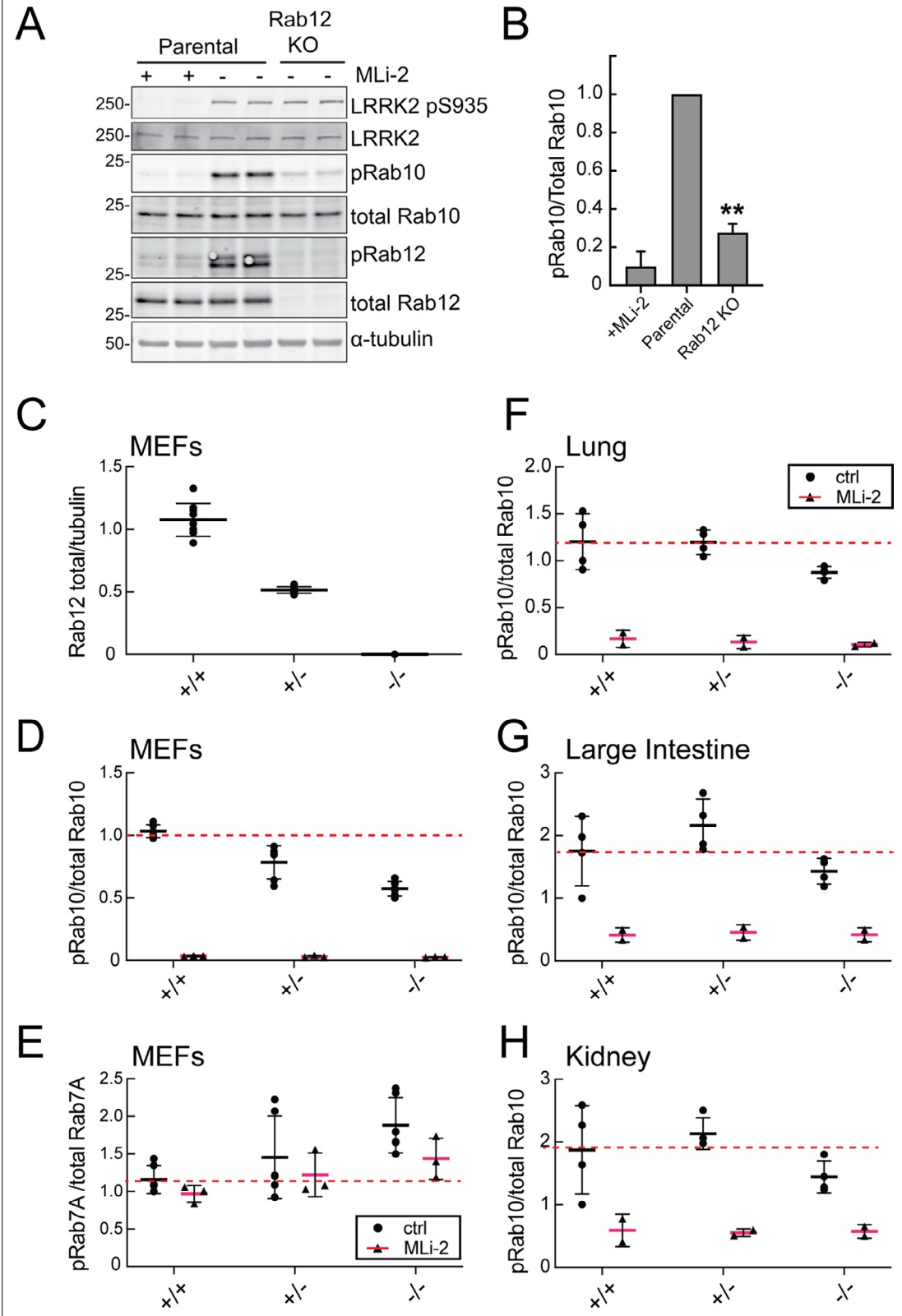

**Figure 2.** Loss of Rab12 decreases phosphoRab10. (**A, B**) Loss of Rab12 decreases phosphoRab10. (**A**) Immunoblot analysis of NIH-3T3-Cas9 cells expressing Rab12 sgRNA (Rab12 KO) or parental cells, +/-MLi2 (200 nM for 2 hr) as indicated. (**B**) Quantitation of phosphoRab10 normalized to total Rab10 from immunoblots in (**A**). Error bars indicate SEM from two experiments carried out in duplicate. **p=0.002 by Student's *t*-test. (**C–H**) Effect of Rab12 knockout on endogenous LRRK2 activity in mouse embryonic fibroblasts (MEFs) (**C–E**) and tissues (**F-H**) derived from Rab12 knockout mice as

*Figure 2 continued on next page*

*Figure 2 continued*

assessed by immunoblot analysis. The quantitation of phosphorylated Rab10 from immunoblots shown in *Figure 2—figure supplements 1 and 2* normalized to respective total Rab10 levels is shown. Quantitation of the phosphorylated Rab7A normalized to respective total Rab7A levels, and total levels of Rab12 are also shown. MLi-2 was administered to MEFs at 100 nM for 1 hr and to mice at 30 mg/kg for 2 hr.

The online version of this article includes the following source data and figure supplement(s) for figure 2:

**Source data 1.** Raw/annotated gels for *Figure 2*.

**Figure supplement 1.** Immunoblots of mouse embryonic fibroblast (MEF) samples in support of *Figure 2*.

**Figure supplement 1—source data 1.** Raw/annotated gels for *Figure 2—figure supplement 1*.

**Figure supplement 2.** Immunoblots of tissue samples in support of *Figure 2*.

**Figure supplement 2—source data 1.** Raw/annotated gels for *Figure 2—figure supplement 2*.

*2019*; *Malik et al., 2021*), appeared to increase moderately as a function of Rab12 loss (*Figure 2E*). Various tissues were analyzed for phosphoRab10 changes in LRRK2 heterozygous and homozygous knockout animals. As shown in *Figure 2F–H*, decreases in phosphoRab10 were detected in the homozygous mouse lung with smaller trends in the large intestine and kidney. Together, these data confirm a role for Rab12 in the LRRK2 signaling pathway that is distinct from that of the previously studied Rab29 protein. We were not able to monitor loss of phosphoRab10 in the brain as phosphoRab10 is more difficult to detect in brain tissue that is enriched in the Rab-specific PPM1H phosphatase (*Berndsen et al., 2019*). Future work will evaluate the consequences of Rab12 knockout in mouse brain and other organs.

## Rab12 overexpression enhances LRRK2 activity

Since loss of Rab12 decreased phosphoRab10 levels, we reasoned that increasing Rab12 should increase phosphoRab10 levels. Indeed, overexpression of GFP-Rab12 in A549 cells led to a tenfold increase in phosphoRab10 levels without changing the levels of LRRK2, PPM1H phosphatase (*Berndsen et al., 2019*) or total Rab10 (*Figure 3A and B*). The ability of Rab12 to activate LRRK2 was specific for that GTPase in that exogenous expression of GFP-tagged Rab8A, Rab10, or Rab29 failed to show the same high level of phosphoRab10 increase – Rab29 yielded about a fivefold enhancement while Rab12 was almost twice as effective in HEK293T cells (*Figure 3C and D*).

The most common, pathogenic, human LRRK2 mutation is LRRK2 G2019S that displays about twofold higher kinase activity than wild type LRRK2; the R1441C mutation activates kinase activity in cells about threefold (*Steger et al., 2016*). Cells expressing each of these forms showed increased phosphorylation upon Rab12 expression (*Figure 3E and F*). It is important to note that Rab12 is a more abundant Rab in most tissues than Rab29; for example, A549 cells contain ~134,000 Rab12 molecules and 25,000 Rab29 molecules per cell. This compares with 5000 copies of LRRK2 and 2.5 million copies of Rab10 (https://copica.proteo.info/#/home). Nevertheless, activation was tested at comparable levels of each Rab protein as monitored using anti-GFP antibodies (*Figure 3C*).

Rab12 activation of LRRK2 did not require Rab12 phosphorylation as the non-phosphorylatable Rab12 S106A was still capable of activation and a phosphomimetic Rab12 S106E failed to increase LRRK2 phospho S1292 (*Figure 3G and H*). Phosphorylation state Rab mutants must be used with great caution as we have shown previously that Rab8A and Rab10 TA mutants fail to correctly localize and the TE mutants bind phosphoRab effectors with much lower affinity than their correctly phosphorylated counterparts (*Dhekne et al., 2018*). Nevertheless, the Rab12 S106A mutant was fully capable of LRRK2 activation.

Similar LRRK2 activation results were obtained using immunofluorescence microscopy to assay phosphoRab10 abundance (*Figure 4*). The phosphoRab10 generated was present on perinuclear membrane compartments (*Figure 4A*) as seen previously by many groups (*Dhekne et al., 2018*; *Dhekne et al., 2021*; *Lara Ordónez et al., 2019*). PhosphoRab10 staining disappeared in cells expressing PPM1H but not in cells expressing the catalytically inactive H153D PPM1H (*Figure 4A and B*). These data were confirmed by immunoblot (*Figure 4C and D*) and suggest that Rab12 is activating LRRK2 along the same pathway of protein phosphorylation studied previously to date.

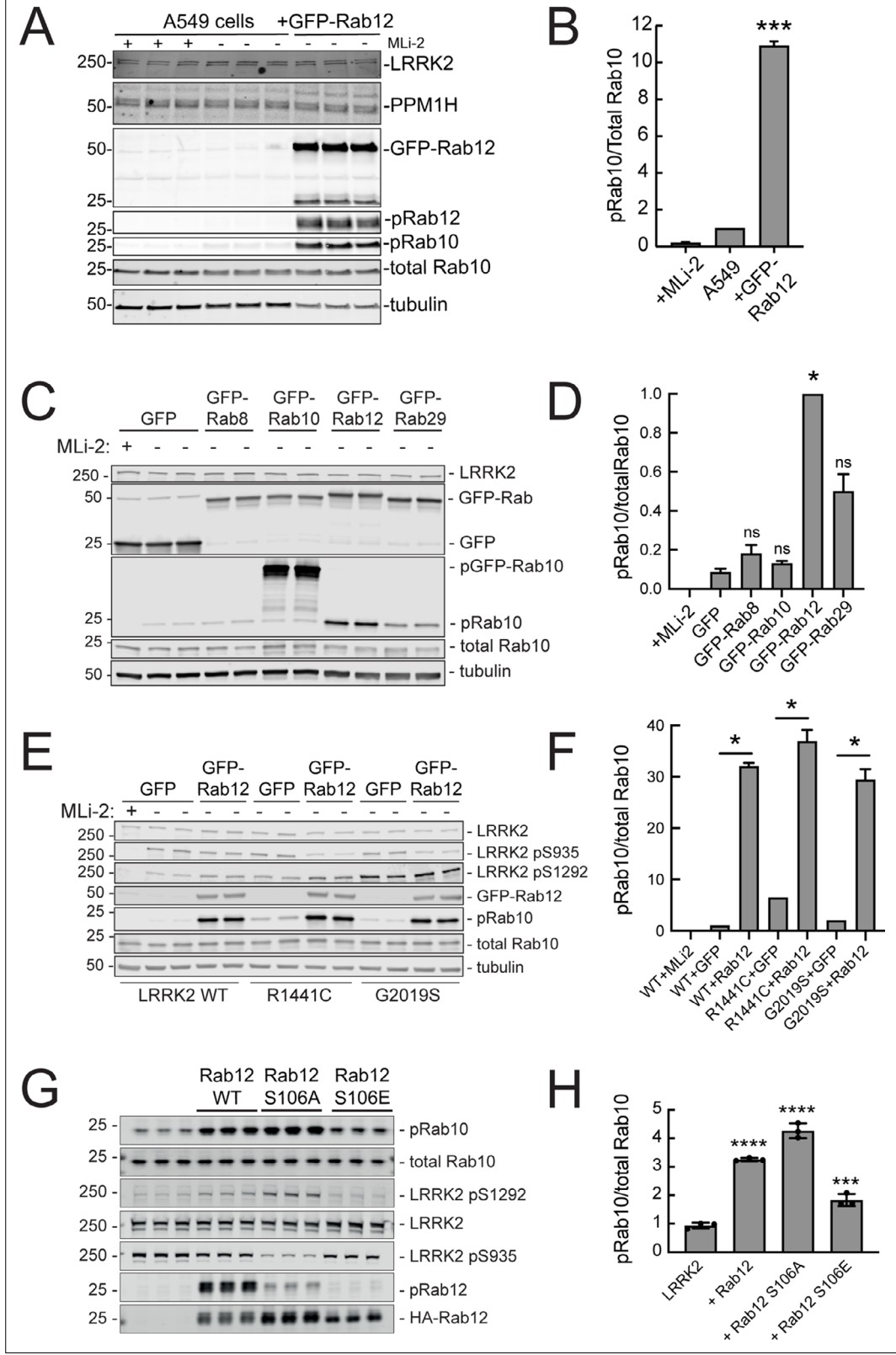

**Figure 3.** Exogenous Rab12 expression increases phosphoRab10 levels in A549 cells. (**A**) Immunoblot analyses of A549 cells stably overexpressing GFP-Rab12; +/-MLi-2 (200 nM for 2 hr) as indicated. (**B**) Quantitation of phosphorylated Rab10 from immunoblots as in (**A**) normalized to total Rab10 levels; error bars indicate SEM from two experiments (\*\*\*p=0.0003 by Student's *t*-test). (**C**) Immunoblot analysis of 293T cells transfected with

*Figure 3 continued on next page*

*Figure 3 continued*

LRRK2 R1441C and GFP, GFP-Rab8, GFP-Rab10, GFP-Rab12, or GFP-Rab29 for 36 hr; +/-MLi2 (200 nM for 2 hr) as indicated. (**D**) Quantitation of phosphorylated Rab10 from immunoblots as in (**C**) normalized to total Rab10 levels. Error bars indicate SEM from two independent experiments; ***p=0.0004 for GFP and GFP-Rab12, *p=0.04 for GFP and GFP-Rab29 with Student's *t*-test. (**E**) Immunoblot analysis of 293T cells transfected with LRRK2 WT, R1441C or G2019S and GFP or GFP-Rab12 for 36 hr, +/-MLi2 (200 nM for 2 hr) as indicated. (**F**) Quantitation of phosphorylated Rab10 from immunoblots as in (**E**) normalized to respective total Rab10 levels. Error bars indicate SEM from two independent experiments; ***p=0.0004 for LRRK2 WT GFP and GFP-Rab12, **p=0.005 for LRRK2 R1441C GFP and GFP-Rab12, **p=0.005 for LRRK2 G2019S GFP and GFP-Rab12 by Student's *t*-test. (**G**) Immunoblot analysis of HEK293 cells expressing wild type FLAG-tagged LRRK2 and the indicated HA-tagged Rab12 constructs. (**H**) Quantitation of phosphorylated Rab10 from immunoblots as in (**G**) normalized to total Rab10; Error bars indicate mean with SD from three independent replicate experiments; ****p<0.0001 for Rab12 WT and Rab12 S106A, ***p=0.0007 for Rab12 S106E by one-way ANOVA relative to LRRK2.

The online version of this article includes the following source data for figure 3:

**Source data 1.** Raw/annotated gels for *Figure 3*.

## Requirements for Rab12 activation of the LRRK2 pathway

It was possible that Rab12 activated a kinase other than LRRK2 to increase Rab10 phosphorylation. This appears not to be the case as GFP-Rab12 expression enhancement of phosphoRab10 levels was not seen in A549 cells lacking LRRK2 expression (*Figure 5A and B*). It was possible that exogenous GFP-Rab12 inhibited overall Rab phosphatase activity, leading to an apparent increase in phosphoRab10 levels. This was also ruled out as cells lacking PPM1H displayed full Rab12-induced enhancement of phosphoRab10 levels (*Figure 5C and D*), about fivefold with or without PPM1H.

## Rab12 expression influences primary ciliogenesis

We showed previously that increased Rab GTPase phosphorylation blocks the formation of primary cilia in cell culture and in specific cell types in the brain (*Steger et al., 2017*; *Dhekne et al., 2018*; *Sobu et al., 2021*). Loss of cilia in cell culture requires Rab10 phosphorylation and its binding to RILPL1 protein (*Dhekne et al., 2018*). If Rab12 expression increases Rab phosphorylation, it would be expected to interfere with primary cilia formation. We tested this in RPE cells that are well ciliated in culture. As shown in *Figure 5E*, overexpression of GFP-Rab12 decreased the percent of RPE cells bearing cilia, after 24 hr of serum starvation to trigger cilia formation. Moreover, knockout of Rab12 from A549 cells that poorly ciliate and only ciliate when plated to full confluency, increased the percentage of ciliated cells upon serum starvation, consistent with a decrease in phosphoRab10 (*Figure 5F*). These experiments show that Rab12 levels regulate primary ciliogenesis downstream of LRRK2 Rab phosphorylation.

## Rab12 activation requires a novel Rab binding site in the LRRK2 Armadillo domain

Previous work has identified specific residues within the LRRK2 Armadillo domain that enable LRRK2 to be recruited to the Golgi by exogenously overexpressed Rab29; these residues support direct Rab29 binding (*McGrath et al., 2021*; *Vides et al., 2022*; *Zhu et al., 2022*). In particular, R361, R399, and K439 contribute to a Rab binding 'Site #1' that supports binding to purified Rab29 ($K_D$ = 1.6 μM; *Vides et al., 2022*; *Figure 6*). Rab8A binds this LRRK2 350–550 region with a similar affinity (2.3 μM) but Rab10 binds less well (5.1 μM; *Vides et al., 2022*). A second site at LRRK2's N-terminus (Site #2, K17/K18) mediates interaction with phosphorylated Rab8A and Rab10 proteins. Rab GTPase binding to either or both sites contributes to LRRK2 membrane association as Rabs are themselves membrane anchored by two covalently attached, 20 carbon geranylgeranyl groups.

AlphaFold (*Jumper et al., 2021*) in conjunction with Colabfold in ChimeraX (*Mirdita et al., 2022*; *Pettersen et al., 2004*) revealed a third Rab binding site (Site #3) when Armadillo domain residues (1–550) were modeled together with Rab12 (*Figure 6B* and *Figure 6—figure supplement 1*). (The Armadillo domain is comprised of residues 1–705; we modeled 1–550 as that portion is biochemically stable and well suited for binding experiments.) The predicted local distance difference test (pLDDT) score (0–100) is a per-residue confidence score, with values greater than 90 indicating high

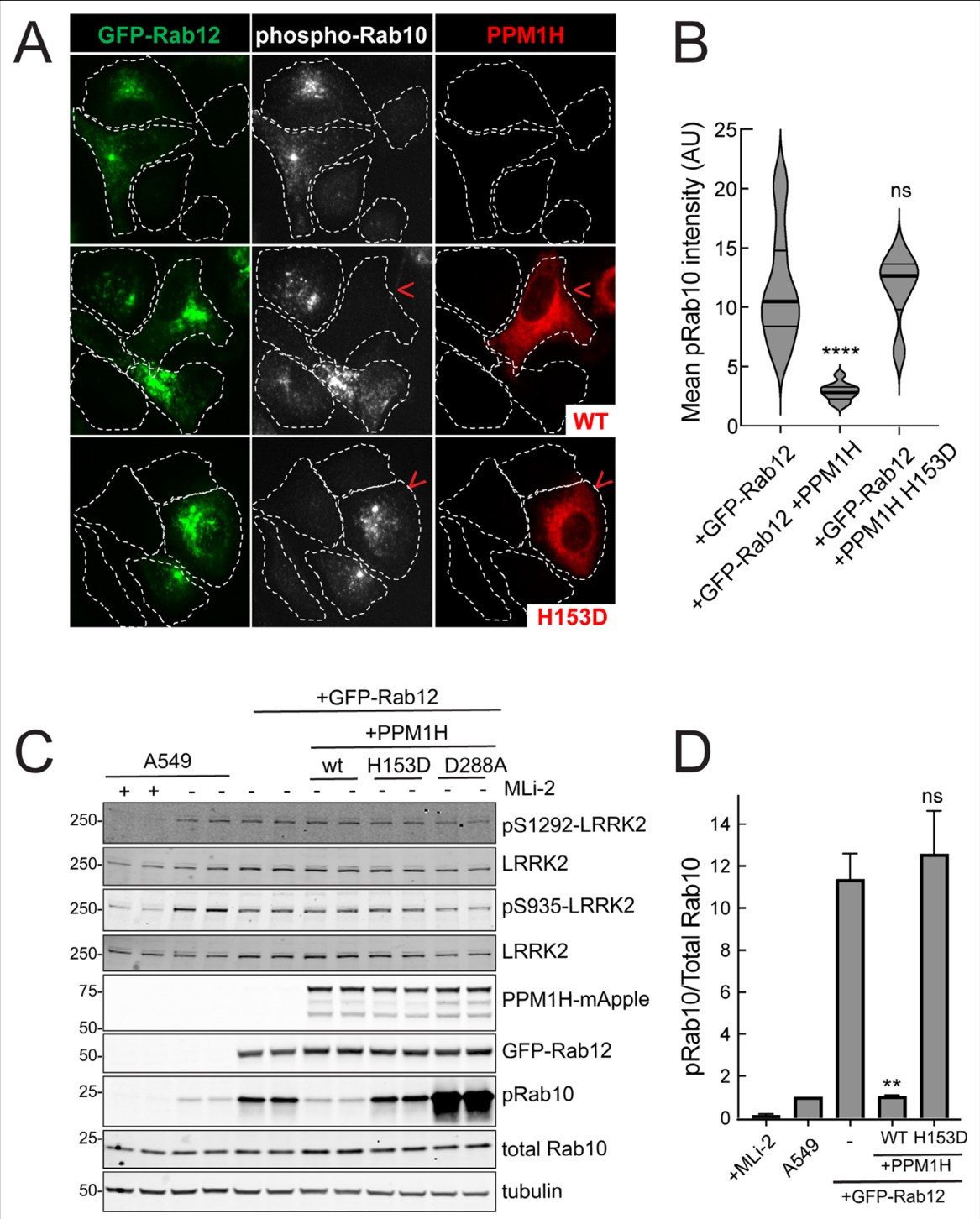

**Figure 4.** PPM1H phosphatase counters phosphoRab10 generated upon Rab12 activation. (**A**) A549 cells stably expressing GFP-Rab12 and PPM1H-mApple (wild type and H153D catalytically inactive mutant) were co-cultured with parental wild type A549 cells on coverslips. PhosphoRab10 was detected by immunofluorescence using rabbit anti-phosphoRab10. Red arrowheads indicate a cell with both GFP-Rab12 and wtPPM1H-mApple or PPM1H H153D. Scale bar = 10μm. (**B**) Quantitation of mean phosphoRab10 fluorescence intensity per cell (Arbitrary units, AU) is shown in the violin plot. Error bars indicate SEM from two independent experiments. At least 10 cells per condition were counted. ****p<0.0001 for GFP-Rab12 and GFP-Rab12+wtPPM1H, ns p=0.9944 for GFP-Rab12 and GFP-Rab12+H153D PPM1H by Student's *t*-test. (**C**) Immunoblot analysis of parental A549 cells or A549 cells stably expressing GFP-Rab12 together with either wtPPM1H, H153D-PPM1H or D288A-PPM1H; +/-MLi2 (200 nM for 2 hr) as indicated. (**D**) Quantitation of phosphorylated Rab10 from immunoblots as in (**A**) normalized to respective total Rab10 levels. Error bars indicate SEM from two

*Figure 4 continued on next page*

*Figure 4 continued*

independent experiments; **p=0.007 for GFP-Rab12 and GFP-Rab12+wtPPM1H, ns p=0.5510 for GFP-Rab12 and GFP-Rab12+H153D-PPM1H by Student's *t*-test.

The online version of this article includes the following source data for figure 4:

**Source data 1.** Raw/annotated gels for *Figure 4*.

confidence; the top 5 structure models (*Figure 6—figure supplement 1*) yielded pLDDT scores of 87.6, 87.5, 86.8, 87.4, and 86.4, respectively, consistent with high-accuracy modeling.

Mutagenesis across this putative Site #3 binding interface yielded full-length LRRK2 proteins with decreased overall activity as monitored by phosphoRab10 levels in HEK293 cells expressing the mutant proteins (*Figure 7A* and *Figure 7—figure supplement 1*). Note that in these experiments the cells rely only on endogenous Rab12 protein. Mutation of E240 and S244 had the greatest impact on LRRK2 activity; remarkably, mutation of F283 to A increased kinase activity twofold. These data demonstrate that Site #3 sequences are important for overall LRRK2 activity.

Mutation of LRRK2 Site #3 E240R and S244R predicted to be important for Rab12 binding blocked the ability of exogenous Rab12 to enhance phosphoRab10 levels (*Figure 7B* and *Figure 7—figure supplement 1*). Moreover, F283A LRRK2 had twofold higher basal activity but was not activated by exogenous Rab12 significantly more than wild type LRRK2 protein. These data strongly suggest that Rab12 activates LRRK2 by binding to Site #3 within the Armadillo domain.

Extensive previous mutagenesis defined Site #1 as being critical for exogenous Rab29-dependent relocalization of LRRK2 to the Golgi complex and apparent activation (*Vides et al., 2022*). It was therefore important to assess whether Rab29's ability to increase phosphoRab10 levels upon overexpression relies upon Site #3. As expected, exogenous expression of Rab29 increased phosphoRab10 levels (albeit to a lower extent than exogenous Rab12 expression; *Figure 3C and D*). However, mutation of Site #3 residues critical for Rab12-mediated LRRK2 activation (E240 and S244) had no effect on the ability of Rab29 to activate LRRK2 kinase (*Figure 7C*). Similarly, mutation of Site #1 residues preferentially decreased the ability of Rab29 to activate LRRK2 with little if any change in Rab12 activation (*Figure 7D*). These experiments show that Rab29 interacts preferentially with Site #1 and demonstrate the Rab12 selectivity of Site #3 for LRRK2 activation.

## Rab12 binds LRRK2 Site #3 directly

These experiments strongly suggest that Rab29 and Rab12 activate LRRK2 by two different routes: Rab29 via binding to LRRK2 Site #1 and Rab12 via binding to Site #3. We validated Rab12 direct binding to Site #3 using purified Rab12 and Armadillo domain proteins mutated at either Site #1 (K439E) or Site #3 (E240R). As shown in *Figure 8*, Rab12 bound as well to the wild type Armadillo domain (*Figure 8A*, 1.4 µM) as to an Armadillo domain construct bearing a Site #1 mutation (*Figure 8B*, 1.6 µM) as determined by microscale thermophoresis. In contrast, the Site #3 E240R mutation abolished the interaction, yielding a $K_D$ of >40 µM (*Figure 8C*). Thus, Rab12 binds tightly and directly to Site #3 in vitro and does not appear to interact with Site #1. Interestingly, the LRRK2 Site #3 F283A mutation that increases kinase activity in cells did not influence Rab12 binding significantly, displaying a $K_D$ of 1.2 µM (*Figure 8D*).

Binding of Rab12 to LRRK2 Site #3 was also detected in cell extracts in co-immunoprecipitation experiments. As shown in *Figure 8E and F*, HA-tagged Rab12 and endogenous Rab12 proteins co-precipitated with FLAG-LRRK2 upon transfection in HEK293T cells. In contrast, significantly less co-precipitation was seen with LRRK2 Site #3 mutant E240R and S244R proteins, with or without exogenous HA-Rab12 expression. Rab12 bound F283A LRRK2 as well as wild type LRRK2 protein, consistent with its binding affinity in vitro.

## PhosphoRab binding is distinct from the Rab12 pathway of LRRK2 activation

We showed previously that phosphoRab binding to Rab binding Site #2 is critical for cooperative LRRK2 membrane recruitment and apparent activation (*Vides et al., 2022*). Thus, it was important to investigate whether Rab12 acts via this feed-forward process. If true, such activation would be predicted to rely on LRRK2 Lys17 and Lys18. As shown in *Figure 8G*, mutation of Lys17 and 18 had

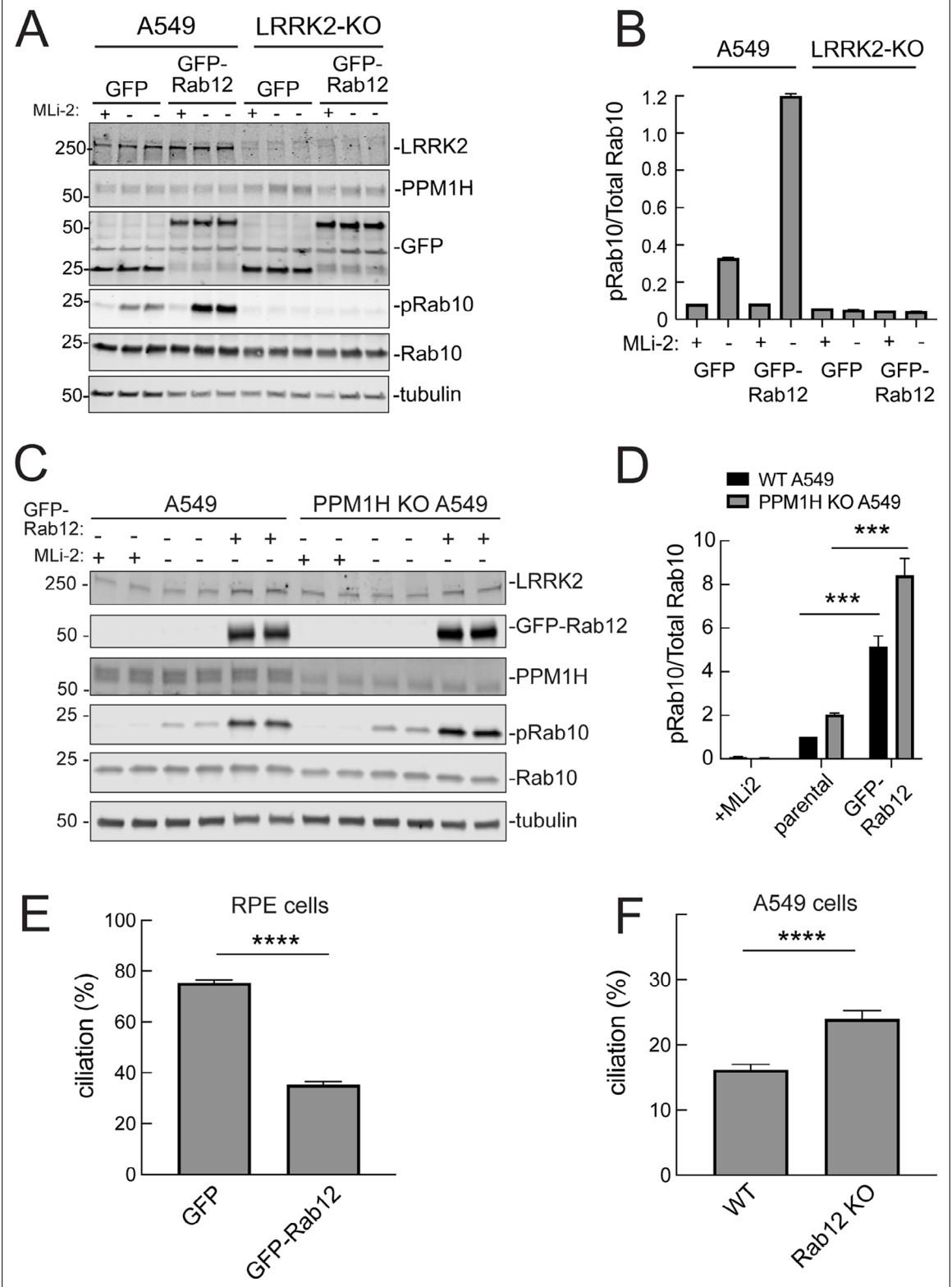

**Figure 5.** Roles of LRRK2 and PPM1H in Rab12 activation of LRRK2. (**A**) Immunoblot analysis of WT and LRRK2 KO A549 cells stably expressing GFP or GFP-Rab12;+/-MLi-2 (200 nM for 2 hr) as indicated. (**B**) Quantitation of phosphorylated Rab10 from immunoblots as in (**A**) normalized to respective total Rab10 levels. (**C**) Immunoblot analysis of WT and PPM1H KO A549 parental cells or cells stably expressing GFP-Rab12;+/-MLi-2 (200 nM for 2 hr) as indicated. (**D**) Quantitation of phosphorylated Rab10 from immunoblots as in (**C**) normalized to respective total Rab10, normalized to WT parental. Error bars indicate SEM from four independent experiments; ***p=0.0002 for both WT and PPM1H KO parental and GFP-Rab12 by Student's *t*-test. (**E**) RPE

*Figure 5 continued on next page*

*Figure 5 continued*

cells stably overexpressing either GFP or GFP-Rab12 were serum starved for 24 hr to trigger ciliation. Cilia were detected using anti-Arl13b antibody and ciliation percentage was calculated by the number of cilia (by Arl13b) per cell (by DAPI). Error bars represent SEM from two independent experiments, >500 cells counted each. ****p<0.0001 by Student's *t*-test. (**F**) WT or Rab12 KO A549 were plated at full confluency and serum starved for 24 hr to trigger ciliation. Percentage of ciliated cells was determined as in (**E**). ****p<0.0001 by Student's *t*-test. Error bars represent SEM from two independent experiments, >500 cells counted each.

The online version of this article includes the following source data for figure 5:

**Source data 1.** Raw/annotated gels for *Figure 5*.

no effect on the ability of Rab12 to increase phosphoRab10 levels in HEK293T cells co-expressing exogenous LRRK2 and GFP-Rab proteins. Once again, Rab12 activation was dramatic and K17/K18 containing-LRRK2 was activated to the same overall level as the K17A/K18A mutant LRRK2 protein. These data are consistent with our finding that non-phosphorylatable Rab12 S106A is still capable of LRRK2 activation (*Figure 3G and H*).

## Rab12 drives LRRK2 activation upon lysosomal or ionophore-triggered stress

As mentioned earlier, under conditions of lysosomal damage, LRRK2 is recruited to lysosomes and participates in the repair of damaged endomembranes (*Eguchi et al., 2018*; *Herbst et al., 2020*; *Bonet-Ponce et al., 2020*). Such stress greatly increases LRRK2 kinase activity (*Kalogeropulou et al., 2020*). *Figure 9A–C* show that Rab12 is required for the modest increase in LRRK2 activity seen upon lysosomal damage triggered by 1 mM LLOME addition for 2 hr in NIH-3T3 cells. In MEFs (*Figure 9D and E*), loss of Rab12 dampened but did not abolish the increase in phosphoRab10 levels, especially at later times. Upon treatment of NIH-3T3 cells with nigericin that also causes mitochondrial stress and is a potent activator of the NLRP3 inflammasome (*Figure 9F–H*), Rab12 knockout diminished Rab10 phosphorylation to control levels. These findings point to the contribution of Rab12 in regulating LRRK2 activity in lysosome repair.

## Discussion

Using an unbiased, genome-wide screen, we have discovered an important and unanticipated role for the understudied Rab12 GTPase in LRRK2 kinase regulation. Loss of Rab12 from NIH-3T3 and MEF cells (and possibly also mouse lung tissue) significantly decreased phosphoRab10 levels, and Rab12 overexpression increased phosphoRab10 levels. The phosphoRab10 increase was LRRK2-dependent, Rab12-specific, and seen with both wild type and pathogenic mutant LRRK2 proteins. PhosphoRab10 showed the same subcellular localization seen in prior work with cells expressing hyperactive LRRK2 proteins and was sensitive to the Rab-specific, PPM1H phosphatase, consistent with Rab12 activation being part of the normal LRRK2 phosphorylation pathway. Moreover, the increased phosphoRab10 generated as a consequence of Rab12-mediated LRRK2 activation influenced primary cilia formation as expected for typical LRRK2 activation. Site-directed mutagenesis in conjunction with computational modeling revealed a new Rab binding site (Site #3) within the LRRK2 Armadillo domain that is needed for Rab12 binding and activation and is not engaged by Rab29 to trigger apparent kinase activation.

*Figure 6* summarizes our current knowledge of Rab GTPase Armadillo domain interactions. Rab29 and its relatives, Rab32 and Rab38, can bind to Site #1 that includes LRRK2 R361, R399, L403, and K439 residues (*McGrath et al., 2021*; *Vides et al., 2022*; *Zhu et al., 2022*); Rab8A is also able to bind at that location (*Vides et al., 2022*). PhosphoRab8A and phosphoRab10 interact with comparable high affinity with LRRK2 K17/18 at Site #2 (*Vides et al., 2022*). This study reveals a third interaction interface on the opposite face of the Armadillo domain (relative to Site #1) that engages Rab12 GTPase. The cryoEM structures of full-length LRRK2 (*Myasnikov et al., 2021*) or LRRK2 in the presence of Rab29 (*Zhu et al., 2022*) both show an extended and flexible Armadillo domain that extends away from the kinase center and would be available for Rab GTPase engagement.

What are the roles of these multiple Rab binding sites? Site #1 can interact with overexpressed Rab29 protein and bring the mostly cytosolic LRRK2 kinase to the surface of the Golgi complex, which will lead to apparent activation. With regard to membrane anchoring, since loss of Rab29 has no detectable consequence for Rab phosphorylation (*Kalogeropulou et al., 2020*), it seems likely

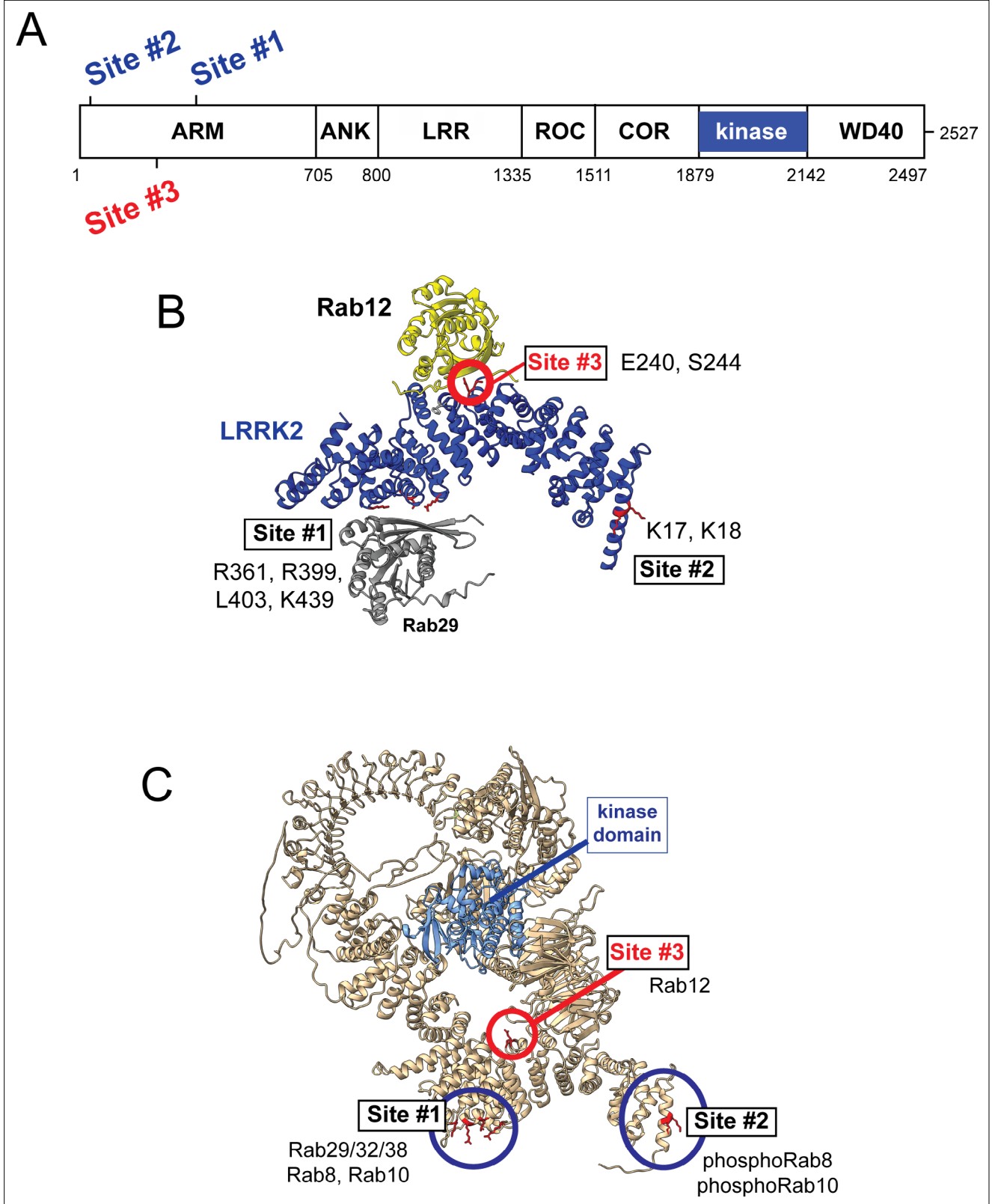

**Figure 6.** Models for Rab interactions with the LRRK2 Armadillo domain. (**A**) Domain organization of LRRK2 with Rab binding sites #1–3 indicated. (**B**) AlphaFold model for LRRK2 Armadillo domain (blue) interaction with Rab12 (yellow) and Rab29 (gray). The Rab12 was docked onto Armadillo using Colabfold in ChimeraX; Rab29 was positioned manually. Site #1 binds Rab29, Site #2 binds phosphorylated Rabs (**Vides et al., 2022**), and Site #3 binds

*Figure 6 continued on next page*

*Figure 6 continued*

Rab12. The key residues for Rab12 binding are circled in red. (C) Full-length AlphaFold model of LRRK2 indicating localization of Rab binding sites; the kinase catalytic domain is highlighted in light blue.

The online version of this article includes the following video and figure supplement(s) for figure 6:

**Figure supplement 1.** Overlay of the top 5 AlphaFold models for Rab12 interaction with the LRRK2 Armadillo domain residues 1–552.

**Figure 6—video 1.** Model of Rab12 (pink) bound to LRRK2 Armadillo domain docked onto the full-length LRRK2 structure.

https://elifesciences.org/articles/87098/figures#fig6video1

that Site #1 can also be occupied by the ubiquitous and more abundant Rab8A or possibly Rab10 GTPases. Site #2 that binds to phosphoRabs will also contribute to the membrane anchoring of LRRK2 kinase (*Vides et al., 2022*); loss of this site decreased overall LRRK2 membrane association at steady state. Site #3 faces the kinase catalytic domain in the AlphaFold model of a putative active LRRK2 protein (*Figure 6* and *Figure 6—video 1*), and we propose that Rab12 binding to Site #3 holds open the kinase to enable substrate access to the active site. *Figure 6—video 1* shows a model of Rab12 (pink) bound to the Armadillo domain overlaid onto the AlphaFold model of full-length LRRK2. This model shows that Rab12 occupancy will push against and clash with sequences adjacent to the kinase catalytic domain (shown in blue); presumably Rab12 binding activates the kinase domain through conformational changes. Given that Rab12's Ser106 phosphorylation site faces the Armadillo domain as part of this Site #3 protein binding interaction, LRRK2 contains at least one additional, yet to be discovered, substrate binding site that positions the Rab phosphorylation site in the correct orientation for LRRK2 kinase phospho-addition.

Rabs 8A, 10, and 12 do not perfectly co-localize in cells yet they can all interact with LRRK2. One possibility is that LRRK2 binds one Rab in each compartment, independently. If Rab8 recruits LRRK2, Rab8 and phosphoRab8 will both cooperate to hold LRRK2 on a Rab8-enriched membrane surface. How would Rab12 come in? It is important to keep in mind the fact that in an A549 cell with 134,000 Rab12 molecules and ~1 million Rab8A proteins, the 5000 LRRK2 molecules may find a subcompartment that contains both Rab8A or 10 and Rab12, despite different primary localizations for the bulk of these Rab proteins. It is also possible that LRRK2 recruited by a Rab to one membrane compartment can phosphorylate a Rab on an adjacent membrane compartment. Future relocalization experiments such as those that anchor LRRK2 on specific subcellular compartments (*Gomez et al., 2019*) may shed important light on this interesting question.

Beyond activating LRRK2, little else is known about Rab12 GTPase function. GFP-Rab12 co-localizes with transferrin receptors and the PAT4 amino acid transporter and depletion of Rab12 increases the levels of both of these proteins, leading Fukuda and colleagues to conclude that it functions in membrane protein delivery from the endocytic recycling compartment to lysosomes (*Matsui and Fukuda, 2011*; *Matsui and Fukuda, 2013*; *Matsui et al., 2011*, *Matsui et al., 2014*). These studies showed further that Rab12 regulates the constitutive degradation of PAT4, indirectly influencing mTORC1 activity by modulating cellular amino acid levels. Later work from McPherson showed that under starvation conditions, the Rab12 guanine nucleotide exchange factor DENND3 is phosphorylated by ULK kinase, enhancing its activity and overall levels of Rab12-GTP (*Xu et al., 2015*). Future work will investigate the consequences of starvation on Rab12 localization and possible roles in autophagy and ciliogenesis regulation. LRRK2 is recruited to damaged lysosomes such as those seen in cells treated with lysosomotropic agents or the LLOME peptide (*Eguchi et al., 2018*; *Herbst et al., 2020*; *Bonet-Ponce et al., 2020*). As we show here, Rab12 also plays a role in activating LRRK2 in that context, but Rab10 phosphorylation was nevertheless seen in Rab12 knockout MEF cells at later times after LLOME addition.

Pathogenic mutations in LRRK2 kinase cause Parkinson's disease, and LRRK2 kinase inhibitors are currently in clinical trials in the hopes of benefiting patients (*Jennings et al., 2022*). This work suggests that small molecules that interfere with Rab12 binding to LRRK2 or other means that decrease Rab12 levels may provide additional avenues to target hyperactive LRRK2 kinase.

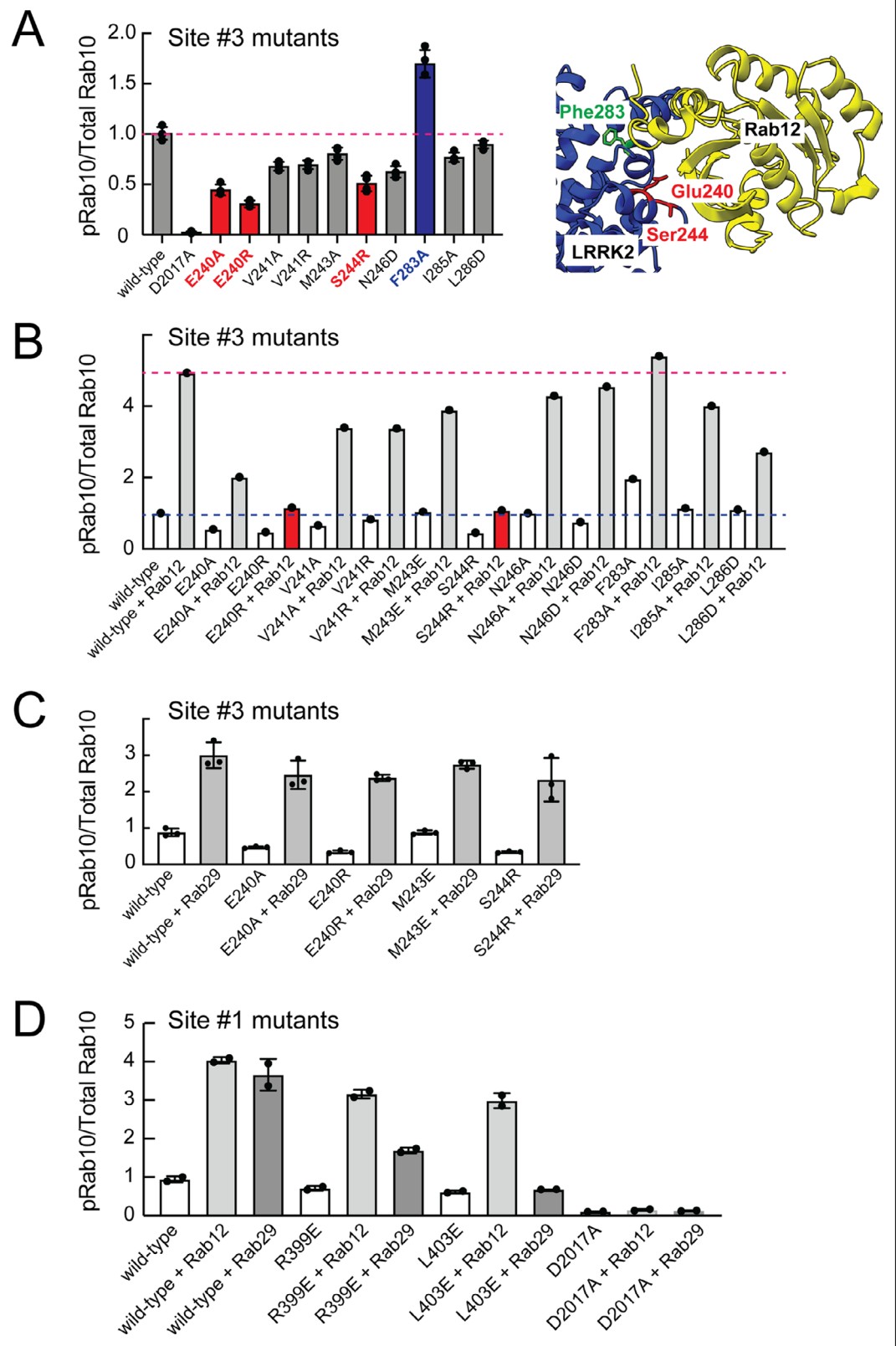

**Figure 7.** Rab binding Site 3 is needed for Rab12- but not Rab29-mediated LRRK2 activation. (**A**) Immunoblot analysis of HEK293 cells transfected with the indicated LRRK2 Site #3 mutants. Shown is quantitation of the fraction of phosphorylated Rab10 from immunoblots as in *Figure 6—figure supplement 1* normalized to respective total Rab10 levels. Shown at right is the structure model for Rab12-ARM domain interaction as in *Figure 6*. (**B**)

*Figure 7 continued on next page*

*Figure 7 continued*

Immunoblot analysis of Site #3 mutants with HA-empty or HA-Rab12 as in (**A**). (**C**) Immunoblot analysis of Site #3 mutants with HA-empty or HA-Rab29 as in (**A**). (**D**) Immunoblot analysis of Site #1 mutants with HA-empty, HA-Rab12, or HA-Rab29 as in (**A**). For all panels, the results from duplicate, independent replicate experiments are shown.

The online version of this article includes the following source data and figure supplement(s) for figure 7:

**Figure supplement 1.** Immunoblots of samples quantified in *Figure 7*.

**Figure supplement 1—source data 1.** Raw/annotated gels for *Figure 7—figure supplement 1*.

## Methods

### Cell culture and Cas9-expressing cell generation

HEK293T, HEK293, NIH-3T3, RPE, A549, and A549 CRISPR knockout lines for LRRK2 and PPM1H (*Berndsen et al., 2019*) were cultured in high-glucose DMEM supplemented with glutamine, sodium pyruvate, and penicillin-streptomycin. All cells were regularly tested for Mycoplasma PCR products using a Lonza Mycoplasma kit. Before the screen, cells were cultured in the presence of plasmocin as prophylaxis against Mycoplasma infection.

Generation of Cas9 expressing NIH-3T3 cells is described in full detail on protocols.io (*Dhekne et al., 2022b*). Briefly, NIH-3T3-Flpin cells were from Thermo Fisher. Early passage cells were transduced with lentivirus carrying HA-Cas9 (Addgene). Cells were selected with blasticidin and single-cell sorted onto a 96-well plate. After 2 wk of culture, 20 individual colonies were picked, expanded, and 5 were analyzed for Cas9 expression and phosphoRab10, LRRK2, and good growth. The two best clones were tested along with a known positive control lentiviral sgRNA, selected with puromycin, and immunoblotted on day 5 to estimate knockout efficiency.

### Validation of genes using pooled knockouts

Two gRNA sequences of each gene to be validated were cloned in pLenti-guide puro vector as described (*Joung et al., 2017*). The plasmids were Sanger sequenced and small-scale lentivirus prepared. NIH-3T3-Cas9 cells were infected with lentiviruses, selected for 3 d, and immediately used for immunofluorescence microscopy or immunoblotting.

### Isolation of Rab12 knockout MEFs

Wild type, heterozygous, and homozygous Rab12 knockout MEFs were isolated from littermate matched mouse embryos at day E12.5 resulting from crosses between heterozygous Rab12 KO/WT mice using the protocol described in *Tonelli, 2023a*. Genotypes were verified via allelic sequencing and immunoblotting analysis. Cells were cultured in DMEM containing 10% (v/v) FBS, 2 mM L-glutamine, penicillin-streptomycin 100 U/mL, 1 mM sodium pyruvate, and 1× non-essential amino acid solution (Life Technologies, Gibco).

### Expanding the sgRNA genome-wide library

The BRIE library from Addgene was expanded according to protocols.io (*Dhekne et al., 2022a*). Briefly, the DNA library was electroporated into Lucigen Endura Duos bacteria and the cells plated onto large format Luria broth agar plates to obtain single colonies across the plate. These plates were grown for 14 hr at 37°C and plasmid extracted using a Machery-Nagel mega-prep kit. Expanded library was PCR amplified using Illumina barcoded PCR primers as described on Addgene and are part of *Supplementary file 1*. PCR products were sequenced with Miseq to confirm uniform distribution of the gRNA sequences across the population. Aliquots of the plasmid library were frozen at –80°C for future use.

### A flow cytometry-based genome-wide screen

The detailed protocols can be found on protocols.io (*Dhekne et al., 2022a* and *Dhekne et al., 2022b*).

Briefly, the screen was performed maintaining a 300× fold representation of guides in the transduced cells (*Pusapati et al., 2018*). For ~79,500 gRNAs, NIH-3T3-Cas9 cells were plated in 20, 15 cm dishes at $5 \times 10^6$ cells per dish. Lentiviral gRNAs were infected at an MOI of 0.2 (for ~100 × $10^6$ cells,

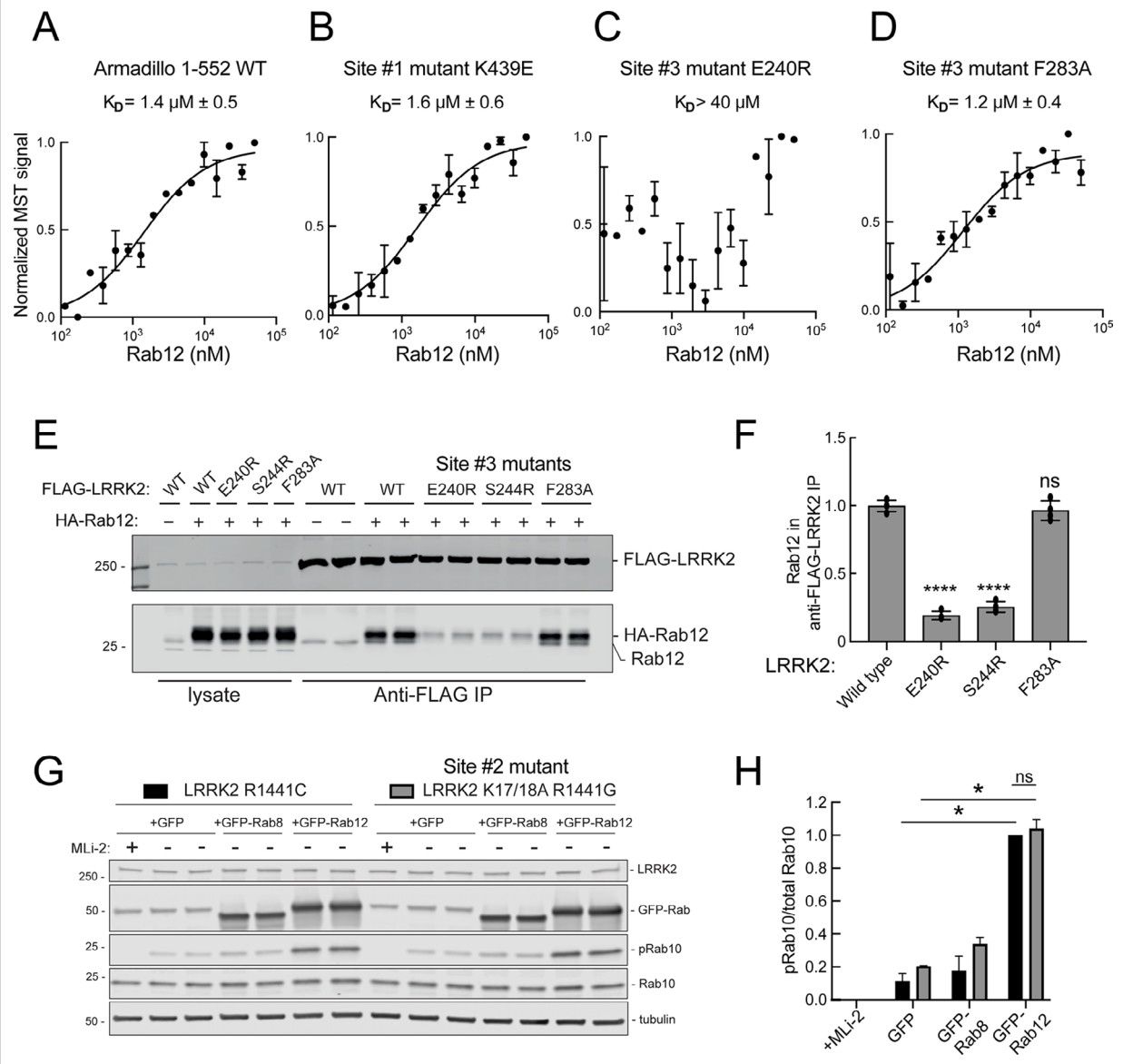

**Figure 8.** Rab12 binds directly to Site #3 and Site #2 is dispensable for Rab12-mediated LRRK2 activation. (**A–D**) Microscale thermophoresis of Rab12 binding to fluorescently labeled LRRK2 Armadillo domain (residues 1–552) wild type (**A**) or bearing the indicated mutations at Site #1: K439E (**B**) or Site #3: E240R (**C**) and F283A (**D**). Purified Rab12 was serially diluted and then NHS-RED-labeled-LRRK2 Armadillo (final concentration 100 nM) was added. Graphs show mean and SEM from two independent measurements, each the average of two replicate runs. (**E**) Immunoblot of anti-FLAG antibody immunoprecipitation of FLAG-LRRK2 wild type or indicated Site #3 mutants with endogenous or co-expressed HA-Rab12 protein in HEK293 cells. Lysate inputs (1.5%) are shown at left; membranes were probed with anti-FLAG or anti-Rab12 antibodies. (**F**) Quantitation of two independent experiments carried out in duplicate as in (**E**). ****$p<0.0001$ for LRRK2 E240R and S244R relative to LRRK2 WT by one-way ANOVA. (**G**) Immunoblot analysis of 293T cells transfected with LRRK2 R1441C or K17/18A R1441G and GFP, GFP-Rab8, or GFP-Rab12 for 36 hr; +/-MLi2 (200 nM for 2 hr). (**H**) Quantitation of the fraction of phosphorylated Rab10 from immunoblots as in (**G**) normalized to respective total Rab10 levels, normalized to LRRK2 R1441C+GFP-Rab12. Error bars indicate SEM from two independent experiments; **$p=0.003$ for LRRK2 R1441C GFP and GFP-Rab12, **$p=0.0044$ for LRRK2 K17/18A R1441G GFP and GFP-Rab12, ns = 0.6 by Student's $t$-test.

The online version of this article includes the following source data for figure 8:

**Source data 1.** Raw/annotated gels for *Figure 8*.

~20 × 10⁶ virus particles). After 48 hr, cells were passed into 60, 15 cm dishes with 1 μg/ml puromycin for selection. After 72 hr, cells in the control plate that did not receive the virus were dead. Puromycin-resistant NIH-3T3-Cas9-BRIE cells were pooled and frozen in cryovial aliquots. Four days before the flow cytometry assay, 40 × 10⁶ cells were thawed and plated into 10, 15 cm dishes and allowed to

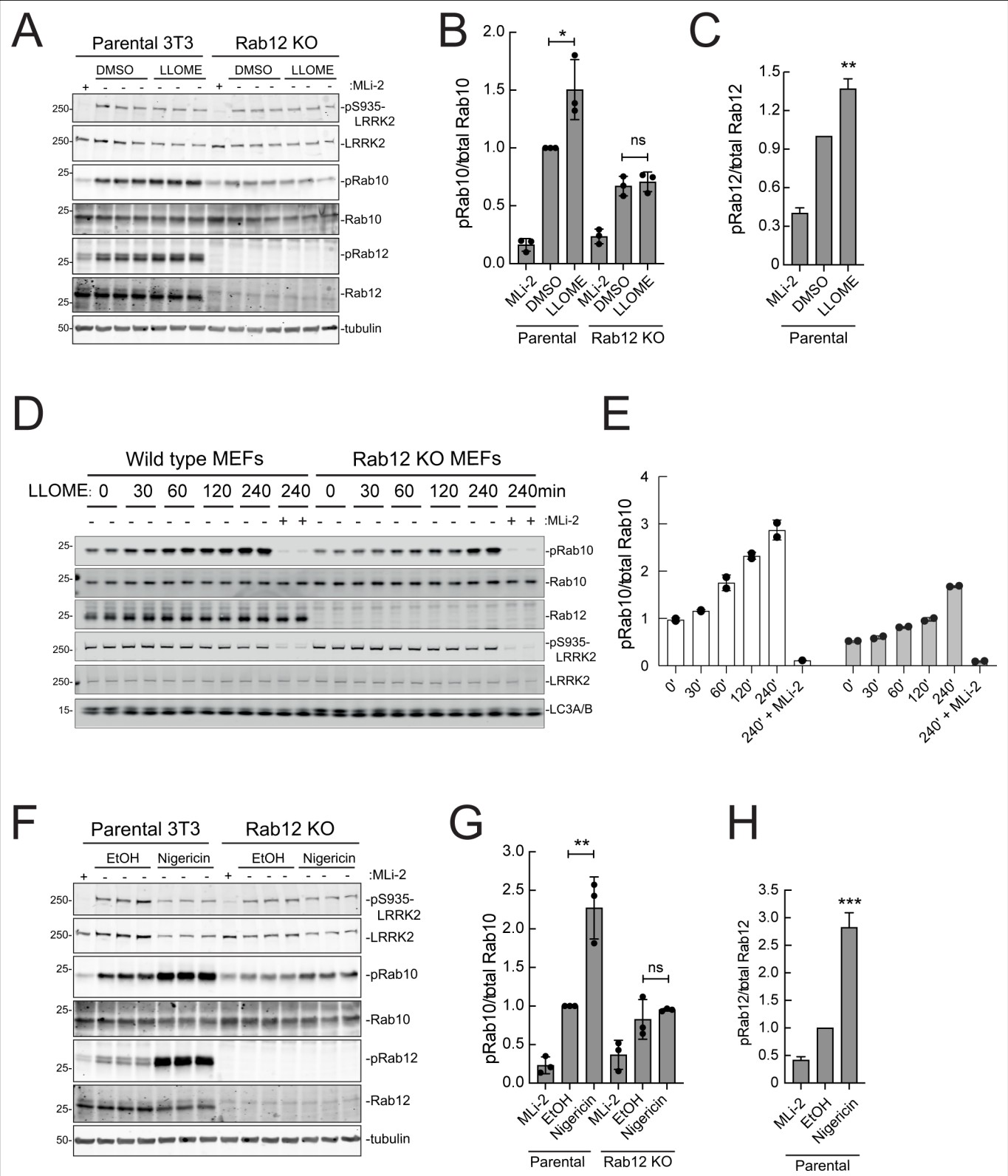

**Figure 9.** Rab12 contributes to LRRK2 activation by LLOME and nigericin. (**A**) Immunoblot analysis of WT and Rab12 KO NIH-3T3 cells treated with 1 mM LLOME for 2 hr,+/-MLi-2 (200 nM for 2 hr) as indicated. (**B**) Quantitation of phosphorylated Rab10 from immunoblots as in (**A**) normalized to total Rab10; Error bars indicate SEM from three experiments. (**C**) Quantitation of phosphorylated Rab12 as in (**A**) normalized to total Rab12; Error bars indicate SEM from three experiments (***p=0.0002 by Student's *t*-test). (**D**) Immunoblot analysis of WT and Rab12 KO MEFs treated with 1 mM LLOME

*Figure 9 continued on next page*

*Figure 9 continued*

for the indicated times, +/-MLi-2 (100 nM for 4 hr) as indicated. (**E**) Quantitation of phosphorylated Rab10 from immunoblots as in (**D**) normalized to total Rab10 levels; error bars indicate mean with SD from two independent replicate experiments. (**F**) Immunoblot analysis of WT and Rab12 KO NIH-3T3 cells treated with 2 µM nigericin for 2 hr, +/-MLi-2 (200 nM for 2 hr) as indicated. (**G**) Quantitation of phosphorylated Rab10 from immunoblots as in (**F**) normalized to total Rab10; error bars indicate SEM from three independent experiments; **p=0.0022 by Student's *t*-test. (**H**) Quantitation of phosphorylated Rab12 from immunoblots as in (**F**) normalized to total Rab12; error bars indicate SEM from three independent experiments; **p=0.0092 by Student's *t*-test.

The online version of this article includes the following source data for figure 9:

**Source data 1.** Raw/annotated gels for *Figure 9*.

attach and grow for 3 d. On the fourth day, cells were trypsinized, resuspended to a cell density of $5 \times 10^6$ cells/mL, passed through a 40 µm cell strainer and fixed with 3% PFA for 30 min, permeabilized with 0.2% Saponin for 30 min, and stained overnight at 4°C with rabbit anti-phosphoRab10 antibody at 1 µg/mL. Cells were then washed and stained with goat anti-rabbit 647 antibody diluted 2 µg/mL for 1 hr at room temperature (RT). Cells were washed, resuspended to $2 \times 10^6$ cells/mL, and injected into a Sony SH800 sorter with FSC of 1, FL4 PMT with a gain of 40%, and sample pressure maintained at level 6. MLi-2-treated and secondary antibody-alone samples were used as negative controls to identify cell population gates. Cells treated with 4 µM nigericin for 3 hr were positive controls for the detection of high level of phosphoRab10.

Cells were sorted based on the histogram of Alexa-647 fluorescent signal. The lowest 5% and highest 7.5% signal containing gates were sorted into two 5 mL collection tubes until each had at least $2 \times 10^6$ cells. To control for total distribution of gRNAs across the population, $10 \times 10^6$ unsorted cells were reserved as input sample. This exercise was performed on two independent sorts from two independent stainings. Sorted cells were pelleted and stored at –80°C for genomic DNA isolation.

## Molecular biology

For genomic DNA extraction, frozen cells were thawed, uncrosslinked, and genomic DNA (gDNA) extracted according to *Dhekne et al., 2022c*. All primers used for PCR amplification for next-generation sequencing (NGS) were ordered as Polypak cartridges purified from the Protein and Nucleic Acid facility, Stanford University. Those used for cloning were ordered unpurified. Primer sequences can be found in *Supplementary file 1*.

Variable sequences were incorporated into forward primer sequences to improve diversity in the NGS run, and eight such primers were pooled in equimolar ratio (Addgene-P5-F [0–8]). Reverse primers were incorporated with TrueSeq indices. PCR was performed as described in protocols.io (*Dhekne et al., 2022a*). Briefly, input plasmid library and each of the genomic DNA libraries were amplified using Titanium-Taq polymerase. PCR products were cleaned up and size selected using Ampure magnetic beads and concentrated by eluting in small volume, quantified with Qubit high-sensitivity dsDNA assay, and finally amplicon size confirmed on an Agilent Bioanalyzer. Each PCR amplicon library (two replicates each of unsorted, low phosphoRab10 and high phosphoRab10) was mixed at equimolar ratio and sequenced at Novogene Co, CA, using their 150 × 2 HiSeq platform.

## Analyses and visualization of NGS data

Raw sequencing reads were mapped to sgRNA sequence guides in the BRIE library using a modified version of count_spacer.py script (*Joung and Gootenberg, 2016*; *Joung et al., 2017*), which provided the count of each sgRNA in each sample. For quality control, evenness of the sgRNA representation was visually assessed by plotting the cumulative distribution of sgRNA representation and quantified using the Gini Index. All samples had a Gini Index lower than 0.42. Consistency between replicates was measured using the Spearman correlation of the sgRNA counts. These quality metrics were computed using Python in a Jupyter Notebook available on GitHub (*Limouse, 2023*).

### sgRNA effect size estimation

The screen data were analyzed using the MAGeCK MLE algorithm (*Li et al., 2014*). For each gene, MAGeCK MLE collapses the effects of individual sgRNAs into a single-gene-level effect size (β-score) and p-value, which quantify the gene contribution to Rab10 phosphorylation in either the positive

direction (β-score < 0, gene knockout decreases phosphoRab10) or negative direction (β-score > 0, gene knockout increases phosphoRab10). p-Values were corrected for multiple hypothesis testing using the false discovery rate (FDR) method. Genes with an FDR < 0.1 were labeled as either positive regulators (β-score < 0) or negative regulators (β-score > 0). For this analysis, samples corresponding to the high phosphoRab10, low phosphoRab10, and unsorted population were included in the design matrix with effect coefficients of +1, –1, and 0. Thus, the reported beta score captures the tendency of a gene knockout to push the cells in the high phosphoRab10 (β-score > 0) or low phosphoRab10 population (β-score < 0). For effect size normalization, the 1000 non-targeting sgRNAs of the Brie library were used, and p-values were determined using the permutation method with 100 rounds of permutation.

To assay consistency in the effect direction across individual sgRNAs targeting the same positive or negative regulator genes determined by the MLE method, we calculated guide-level log$_2$ fold change in the high GFP population versus low GFP population using the MAGeCK RRA method. For this analysis, sgRNAs with fewer than 100 counts in both the high and low GFP samples were discarded. As with the MLE method, effect sizes were normalized using the log$_2$ fold change distribution of the non-targeting sgRNAs.

The MAGeCK output files were loaded as data frames in R (*R Development Core Team, 2021*) and processed with dplyr and ggplot to generate volcano plots, rank plots, and sgRNA-level log$_2$ fold change plots. Code used to run MAGeCK and generate each figure is available on GitHub (*Limouse, 2023*).

All primers, gRNAs, and screen results are included as *Supplementary file 1*.

## Lentiviral preparation and transduction

Large-scale lentiviral preparation for generating pooled lentiviral gRNA libraries was performed according to a modified protocol from *Joung et al., 2017* and is published on protocols.io (*Dhekne et al., 2022a*). Briefly, low-passage HEK293T cells were transfected with BRIE library along with the packaging plasmids and viral supernatant was collected 48 hr (day 2) and 72 hr (day 3) post-transfection. These two separate days of supernatants were pooled, filtered through 0.45 µm, and frozen at –80°C. An aliquot of the frozen virus was used for titration such that <30% of the cells were transduced and showed puromycin resistance. An estimate of the number of virus particles/µL was made. For small-scale preparations of lentiviruses to express individual gRNAs or GFP-tagged Rab GTPases, a standard lentiviral protocol was used as is published in protocols.io (*Dhekne and Pfeffer, 2022a*).

For individual cell lines, RPE and A549 cells were transduced with the relevant virus (GFP, GFP-Rab12, wtPPM1H-mApple, PPM1H H153D-mApple, PPM1H-D288A mApple) and 5 µg/mL polybrene. After 72 hr, cells were either selected for protein expression with puromycin or sorted for the relevant fluorescent protein expression. Sorted cells were tested for protein expression by immunoblot.

## HEK293 overexpression assays

### Rab specificity of LRRK2 activation upon overexpression

HEK293T cells were seeded into six-well plates and transiently transfected at 60–70% confluency using polyethylenimine (PEI) transfection reagent. 1 µg of Flag-LRRK2 WT, R1441C, K17/18A R1441G, and 0.5 ug of GFP, GFP-Rab8, GFP-Rab10, GFP-Rab12, or GFP-Rab29 and 7.5 ug of PEI were diluted in 200 µL Opti-MEM Reduced serum medium (Gibco) per well. 36 hr after transfection, cells were treated with 200 nM MLi-2 for 2 hr as indicated and lysed in ice-cold lysis buffer. Samples were prepared for immunoblotting analysis as below.

### Activation of LRRK2 Site #3 and Site #1 mutants

HEK293 cells were seeded into six-well plates and transiently transfected at 60–70% confluence using PEI transfection reagent with Flag-LRRK2 wild type or variant plasmids. 2 µg of plasmid and 6 µg of PEI were diluted in 0.5 mL of Opti-MEM Reduced serum medium (Gibco) per single well. For co-over-expression experiments, 1.6 µg of Flag-LRRK2 wild type or variant plasmids, 0.4 µg of HA-Rab12 (wild type or phosphomutants), HA-Rab29 or HA-empty, and 6 µg of PEI were diluted in 0.5 mL of Opti-MEM Reduced serum medium (Gibco) per single well. Cells were lysed 24 hr post-transfection in an ice-cold lysis buffer containing 50 mM Tris–HCl pH 7.4, 1 mM EGTA, 10 mM 2-glycerophosphate,

50 mM sodium fluoride, 5 mM sodium pyrophosphate, 270 mM sucrose, supplemented with 1 µg/mL microcystin-LR, 1 mM sodium orthovanadate, cOmplete EDTA-free protease inhibitor cocktail (Roche), and 1% (v/v) Triton X-100. Lysates were clarified by centrifugation at 15,000 × g at 4°C for 15 min, and supernatants were quantified by Bradford assay. Detailed methods for cell transfection and cell lysis can be found on protocols.io (*Tonelli et al., 2021* and *Purlyte et al., 2022*).

## Co-immunoprecipitation analysis of LRRK2 and Rab12 in HEK293 cells

HEK293 cells were seeded into 10 cm plates and transiently transfected at 70–80% confluence using Lipofectamine 2000 transfection reagent with FLAG-tagged LRRK2 wild type or variant plasmids and HA-Rab12 or HA-empty. Cells were lysed 24 hr post-transfection in ice-cold lysis buffer containing 50 mM Tris-HCl pH 7.4, 150 mM NaCl, 1 mM EGTA, 270 mM sucrose, supplemented with 1× phosSTOP phosphatase inhibitor cocktail (PhosSTOP tablet: Roche, REF# 04906837001), 1× protease inhibitor cocktail (cOmplete EDTA-free protease inhibitor cocktail tablet: Roche, REF# 1187358000) and 0.1% (v/v) NP40-Alternative. 1 mg of whole-cell lysate was used to immunoprecipitate LRRK2 with 25 µL anti-FLAG M2 resin for 1 hr at 4°C. Immunoprecipitates were washed three times with 50 mM Tris–HCl pH 7.4, 150 mM NaCl, and eluted by adding 25 µL of 2× lithium dodecyl sulfate (LDS) loading buffer to the resin. A detailed method can be found on protocols.io (*Tonelli, 2023b*).

## Mice

The Rab12 knockout mouse strain used for this research project, C57BL/6N-Rab12em1(IMPC)J/Mmucd (RRID:MMRRC_049312-UCD) was obtained from the Mutant Mouse Resource and Research Center (MMRRC) at the University of California at Davis and was donated to the MMRRC by the KOMP Repository, University of California, Davis (originating from Stephen Murray, The Jackson Laboratory). Mice selected for this study were maintained under specific pathogen-free conditions at the University of Dundee (UK). All animal studies were ethically reviewed and carried out in accordance with the Animals (Scientific Procedures) Act 1986 and regulations set by the University of Dundee and the U.K. Home Office. Animal studies and breeding were approved by the University of Dundee ethical committee and performed under a U.K. Home Office project license. Mice were housed at an ambient temperature (20–24°C) and humidity (45–55%) and were maintained on a 12 hr light/12 hr dark cycle, with free access to food and water. For the experiments described in *Figure 2* and *Figure 2—figure supplement 1*, 3-month-old littermate or age-matched mice of the indicated genotypes were injected subcutaneously with vehicle (40% [w/v] (2-hydroxypropyl)-β-cyclodextrin; Sigma-Aldrich #332607) or MLi-2 dissolved in the vehicle at a 30 mg/kg final dose. Mice were killed by cervical dislocation 2 hr following treatment, and the collected tissues were rapidly snap frozen in liquid nitrogen.

## Quantitative immunoblotting analysis

### Cells

Quantitative immunoblotting analysis to measure levels of proteins was performed according to the protocol on protocols.io (*Tonelli and Alessi, 2021*). Briefly, cells were lysed in lysis buffer (50 mM Tris–HCl pH 7.4, 1 mM EGTA, 10 mM 2-glycerophosphate, 50 mM sodium fluoride, 5 mM sodium pyrophosphate, 270 mM sucrose, supplemented with 1 µg/mL microcystin-LR, 1 mM sodium orthovanadate, cOmplete EDTA-free protease inhibitor cocktail [Roche], and 1% [v/v] Triton X-100). Lysates were clarified by centrifugation at 15,000 × g at 4°C for 10 min. Protein concentration was measured by Bradford and samples equalized and SDS sample buffer added. Samples were run on 4–20% precast gels (Bio-Rad) and transferred onto nitrocellulose membranes. Membranes were blocked in 5% milk with TBST for 1 hr and incubated with specific primary antibodies overnight at 4°C.

### Tissues

Quantitative immunoblotting analysis to measure levels of Rab10, phosphoRab10, LRRK2, pS935 LRRK2 was performed as described in *Tonelli and Alessi, 2021*. Briefly, snap-frozen tissues were thawed on ice in a tenfold volume excess of ice-cold lysis buffer containing 50 mM Tris–HCl pH 7.4, 1 mM EGTA, 10 mM 2-glycerophosphate, 50 mM sodium fluoride, 5 mM sodium pyrophosphate, 270 mM sucrose, supplemented with 1 µg/mL microcystin-LR, 1 mM sodium orthovanadate, cOmplete EDTA-free protease inhibitor cocktail (Roche), and 1% (v/v) Triton X-100 and homogenized using a Precellys Evolution system, employing three cycles of 20 s homogenization (6800 rpm) with 30 s

intervals. Lysates were centrifuged at 15,000 × $g$ for 30 min at 4°C, and supernatants were collected for subsequent Bradford assay and immunoblot analysis.

For blots, the following primary antibodies were used: mouse anti-total LRRK2 (Neuromab N241A/34), rabbit anti-LRRK2 pS935 (ab133450, Abcam), rabbit anti-LRRK2 pS1292 (ab203181, Abcam), rabbit anti-pT73 Rab10 (ab230261, Abcam), mouse anti-total Rab10 (ab104859, Abcam), rabbit anti-pS106 Rab12 (ab256487, Abcam), rabbit anti-total Rab12 (18843-1-AP, Proteintech), sheep anti-total Rab12 (SA227, MRC Reagents and Services), rabbit anti-pS72 Rab7A (ab302494, Abcam), mouse anti-total Rab7A (R8779, Sigma), rabbit anti-pT71 Rab29 (ab241062, Abcam), mouse anti-alpha tubulin (Cell Signaling Technologies, 3873S), rat anti-HA tag (Cat#11867423001, Roche), sheep anti-PPM1H (DA018, MRC Reagents and Services), anti-DYKDDDDK Tag (D6W5B) rabbit mAb (Cell Signaling Technologies, 14793), and rabbit anti-LC3 A/B (Cell Signaling Technologies, 4108). Primary antibody probes were detected using IRdye labeled 1:10,000 diluted secondary antibodies (goat anti-mouse 680, goat anti-rabbit 800, goat anti-chicken 680, donkey anti-goat 800). Membranes were scanned on the LI-COR Odyssey Dlx scanner. Images were saved as .tif files and analyzed using the gel scanning plugin in ImageJ.

## Immunofluorescence, microscopy, and Image analysis

For individual gene knockout validation by microscopy, NIH-3T3-Cas9 cells were transduced with sgRNA lentiviruses for 48 hr, then selected for 3 d with 1 μg/mL puromycin. On day 6, cells were plated at 30% confluency (75,000 cells) on coverslips in a 24-well plate. After 24 hr, cells were washed and fixed with 3% paraformaldehyde for 30 min at RT, permeabilized with 0.1% Saponin for 30 min, blocked with 2% BSA, and stained with rabbit anti-phosphoRab10 and mouse anti-p115 polyclonal antibody for 2 hr at RT.

A549 cells stably expressing GFP-Rab12 and PPM1H-mApple were co-plated with parental A549 cells on coverslips for 24 hr. Cells were then fixed, stained, and imaged for phosphoT73 Rab10 as described below. Cells were washed and stained with DAPI (0.1 μg/mL), donkey anti-mouse 488, and donkey anti-rabbit 568 (1:2000) for 1 hr at RT. After washing the secondary antibody, coverslips from all wells were mounted on slides using Mowiol. Staining of cells for immunofluorescence is described in the protocol (*Dhekne and Pfeffer, 2022b*). After the coverslips dried, unbiased multi-position images were obtained using a spinning disk confocal microscope (Yokogawa) with an electron multiplying charge coupled device (EMCCD) camera (Andor, UK) and a 100 ×1.4 NA oil immersion objective. Image acquisition was performed using the multidimensional acquisition using Metamorph. All images were analyzed using an automated pipeline built using Cell Profiler. Whole-cell intensities of phosphoRab10 were extracted as median and mean intensities of phosphoRab10 across the cell. Given the non-uniform nature of the phosphoRab10 dispersal inside cells, median intensity across cell was used for plotting graphs. Images histograms were adjusted on Fiji and are presented as maximum intensity projections.

Figures were made in Adobe illustrator. Graphs and statistical analyses were performed in GraphPad Prism.

## LRRK2 Armadillo domain and Rab12 purification

His-Rab12 Q101L, His-LRRK2 Armadillo WT, K439E, E240R, and F283A were purified after expression in *Escherichia coli* BL21 (DE3 pLys). Detailed protocols can be found in *Gomez et al., 2019*, *Gomez et al., 2020*, *Dhekne et al., 2021* and *Vides and Pfeffer, 2021*. Bacterial cells were grown at 37°C in Luria Broth and induced at A600 nm = 0.6–0.7 by the addition of 0.3 mM isopropyl-1-thio-β-d-galactopyranoside (Gold Biotechnology) and harvested after growth for 18 hr at 18°C. The cell pellets were resuspended in ice-cold lysis buffer (50 mM HEPES, pH 8.0, 10% [vol/vol] glycerol, 500 mM NaCl, 10 mM imidazole, 5 mM MgCl$_2$, 0.2 mM tris(2-carboxyethyl) phosphine [TCEP], 20 μM GTP, and EDTA-free protease inhibitor cocktail [Roche]). The resuspended bacteria were lysed by one passage through an Emulsiflex-C5 apparatus (Avestin) at 10,000 lbs/in$^2$ and centrifuged at 40,000 rpm for 45 min at 4°C in a Beckman Ti45 rotor. Cleared lysate was filtered through a 0.2 μm filter (Nalgene) and passed over a HiTrap TALON crude 1 mL column (Cytiva). The column was washed with lysis buffer until absorbance values reached pre-lysate values. Protein was eluted with a gradient from 20 to 500 mM imidazole containing lysis buffer. Peak fractions were analyzed by 4–20% SDS-PAGE to locate protein. The eluate was buffer exchanged and further purified by gel filtration on Superdex-75

(GE Healthcare) with a buffer containing 50 mM HEPES, pH 8, 5% (vol/vol) glycerol, 150 mM NaCl, 5 mM $MgCl_2$, 0.2 mM tris(2-carboxyethyl) phosphine (TCEP), and 20 μM GTP.

## Microscale thermophoresis

A detailed method can be found on protocols.io (*Vides and Pfeffer, 2021*).

Protein–protein interactions were monitored by microscale thermophoresis using a Monolith NT.115 instrument (NanoTemper Technologies). His LRRK2 Armadillo (1–552) WT, K439E, E240R, and F283A were labeled using RED-NHS 2nd Generation (Amine Reactive) Protein Labeling Kit (NanoTemper Technologies). For all experiments, unlabeled Rab12 was titrated against a fixed concentration of the fluorescently labeled LRRK2 Armadillo (100 nM); 16 serially diluted titrations of the unlabeled protein partner were prepared to generate one complete binding isotherm. Binding was carried out in reaction buffer (50 mM HEPES pH 8, 150 mM NaCl, 5 mM $MgCl_2$, 0.2 mM tris(2-carboxyethyl) phosphine [TCEP], 20 μM GTP, 5% [vol/vol] glycerol, 5 μM BSA, 0.01% Triton-X) in 0.5 mL Protein LoBind tubes (Eppendorf) and allowed to incubate in the dark for 30 min before loading into NT.115 premium-treated capillaries (NanoTemper Technologies). A red LED at 20% excitation power (red filter, excitation 605–645 nm, emission 680–685 nm) and IR-laser power at 60% was used for 30 s followed by 5 s of cooling. Data analysis was performed with NTAffinityAnalysis software (NanoTemper Technologies) in which the binding isotherms were derived from the raw fluorescence data and then fitted with both NanoTemper software and GraphPad Prism to determine the Kd using a nonlinear regression method. The binding affinities determined by the two methods were similar. Shown are averaged curves of Rab GTPase-binding partners from two independent experiments, with averaged replicates from each run.

## Acknowledgements

This study was funded by the joint efforts of The Michael J Fox Foundation for Parkinson's Research (MJFF) (MJFF grant no. 009258 to SRP and DRA and 021132 to SRP) and Aligning Science Across Parkinson's (ASAP) initiative. MJFF administers the grant (ASAP-000463, SRP and DRA) on behalf of ASAP and itself. CYC was supported by training grant NIH 5T32 GM007276. Funds were also provided by the Medical Research Council (grant no. MC_UU_00018/1 [DRA]), the pharmaceutical companies supporting the Division of Signal Transduction Therapy Unit Boehringer-Ingelheim, GlaxoSmithKline, Merck KGaA (DRA). For the purpose of open access, the authors have applied a CC-BY public copyright license to the Author Accepted Manuscript version arising from this submission.

We are especially grateful to Drs. Ganesh Puspati and Rajat Rohatgi for critical guidance in performing the NIH-3T3 cell CRISPR screen, Jacqueline Bendrick and Yohan Auguste for help with Figure 8A-C, Dr. Jonas Nikoloff for help with Figure 5E and F, Collin Chiu for help with AlphaFold, and Dr. Sreeja Nair for help sustaining clones while HD recovered from COVID. We also thank the excellent technical support of the MRC Protein Phosphorylation and Ubiquitylation Unit (PPU) DNA sequencing service (coordinated by Gary Hunter), the MRC-PPU tissue culture team (coordinated by Edwin Allen), the MRC-PPU Reagents and Services antibody and protein purification teams (coordinated by Dr James Hastie), and the MRC-PPU Genotyping team (coordinated by Gail Gilmour).

## Additional information

### Funding

| Funder | Grant reference number | Author |
| --- | --- | --- |
| Aligning Science Across Parkinson's | 000463 | Dario R Alessi<br>Suzanne R Pfeffer |
| Michael J. Fox Foundation for Parkinson's Research | 009258 | Dario R Alessi<br>Suzanne R Pfeffer |
| Michael J. Fox Foundation for Parkinson's Research | 021132 | Suzanne R Pfeffer |
| National Institutes of Health | 5T32 GM007276 | Claire Y Chiang |

| Funder | Grant reference number | Author |
|---|---|---|
| Medical Research Council | MC_UU_00018/1 | Dario R Alessi |
| Boehringer Ingelheim | | Dario R Alessi |
| GlaxoSmithKline | | Dario R Alessi |
| Merck KGaA | | Dario R Alessi |

The funders had no role in study design, data collection and interpretation, or the decision to submit the work for publication.

## Author contributions

Herschel S Dhekne, Conceptualization, Data curation, Formal analysis, Validation, Investigation, Methodology, Writing – review and editing; Francesca Tonelli, Conceptualization, Formal analysis, Investigation, Writing – review and editing; Wondwossen M Yeshaw, Conceptualization, Data curation, Formal analysis, Investigation, Visualization, Writing – review and editing; Claire Y Chiang, Conceptualization, Data curation, Formal analysis, Validation, Investigation, Writing – review and editing; Charles Limouse, Data curation, Software, Formal analysis, Visualization, Writing – review and editing; Ebsy Jaimon, Elena Purlyte, Investigation, Writing – review and editing; Dario R Alessi, Conceptualization, Data curation, Supervision, Funding acquisition, Project administration, Writing – review and editing; Suzanne R Pfeffer, Conceptualization, Data curation, Formal analysis, Supervision, Funding acquisition, Visualization, Writing - original draft, Project administration

## Author ORCIDs

Herschel S Dhekne (ID) https://orcid.org/0000-0002-2240-1230
Francesca Tonelli (ID) http://orcid.org/0000-0002-4600-6630
Wondwossen M Yeshaw (ID) http://orcid.org/0000-0002-3134-3458
Claire Y Chiang (ID) https://orcid.org/0000-0002-0999-9856
Charles Limouse (ID) https://orcid.org/0000-0003-2589-4576
Ebsy Jaimon (ID) https://orcid.org/0000-0001-6845-2095
Elena Purlyte (ID) https://orcid.org/0000-0001-7291-1549
Dario R Alessi (ID) http://orcid.org/0000-0002-2140-9185
Suzanne R Pfeffer (ID) http://orcid.org/0000-0002-6462-984X

## Ethics

All animal studies were ethically reviewed and carried out in accordance with the Animals (Scientific Procedures) Act 1986 and regulations set by the University of Dundee and the U.K. Home Office.

## Decision letter and Author response

Decision letter https://doi.org/10.7554/eLife.87098.sa1
Author response https://doi.org/10.7554/eLife.87098.sa2

# Additional files

## Supplementary files

• Supplementary file 1. List of primers, gRNAs, and all screen results.

• MDAR checklist

## Data availability

All primary data associated with each figure has been deposited in a repository and can be found at https://doi.org/10.5281/zenodo.7633917, https://doi.org/10.5281/zenodo.8035447, and https://doi.org/10.5281/zenodo.7659210.

The following datasets were generated:

| Author(s) | Year | Dataset title | Dataset URL | Database and Identifier |
|-----------|------|---------------|-------------|-------------------------|
| Tonelli F | 2023 | Genome wide screen reveals Rab12 GTPase as a critical activator of pathogenic LRRK2 kinase | https://doi.org/10.5281/zenodo.7633917 | Zenodo, 10.5281/zenodo.7633917 |
| Dhekne HS, Tonelli F, Yeshaw WM, Chiang CY, Limouse C, Jaimon E, Purlyte E, Alessi DR, Pfeffer SR | 2023 | Genome-wide screen reveals Rab12 GTPase as a critical activator of Parkinson's disease-linked LRRK2 kinase | https://doi.org/10.5281/zenodo.8035447 | Zenodo, 10.5281/zenodo.8035447 |
| Dhekne HS, Tonelli F, Yeshaw WM, Chiang CY, Limouse C, Jaimon E, Purlyte E, Alessi DR, Pfeffer SR | 2023 | Genome-wide screen reveals Rab12 GTPase as a critical activator of pathogenic LRRK2 kinase | https://doi.org/10.5281/zenodo.7659210 | Zenodo, 10.5281/zenodo.7659210 |

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

# Appendix 1

## Appendix 1—key resources table

| Reagent type (species) or resource | Designation | Source or reference | Identifiers | Additional information |
|---|---|---|---|---|
| Antibody | anti-LRRK2 (mouse monoclonal) | Antibodies Incorporated/ NeuroMab | N241A/34 (RRID:AB_10675136) | 1:1000 |
| Antibody | anti-LRRK2 phospho S935 (rabbit monoclonal) | MRC PPU Reagents and Services, University of Dundee | UDD2 10 (Gulbranson et al., 2017) (RRID:AB_2921228) | 1:1000 |
| Antibody | anti-LRRK2 phospho S1292 (rabbit monoclonal) | Abcam | ab203181 (RRID:AB_2921223) | 1:1000 |
| Antibody | anti-Rab10 (mouse monoclonal) | Nanotools | 0680–100/Rab10-605B11 (RRID:AB_2921226) | 1:1000 |
| Antibody | anti-Rab10 (phospho T73) (rabbit monoclonal) | Abcam | Ab230261 (RRID:AB_2811274) | 1:1000 |
| Antibody | anti-Rab10 (phospho T73 MJFR-21-22-5) (rabbit monoclonal) | Abcam | Ab241060 (RRID:AB_2884876) | 1:1000 |
| Antibody | anti-FLAG M2 (mouse monoclonal) | Millipore Sigma | F-1804 (RRID:AB_262044) | 1:2000 |
| Antibody | anti-DYKDDDDK Tag (D6W5B) (rabbit monoclonal) | Cell Signaling Technology | #14793 (RRID:AB_2572291) | 1:1000 |
| Antibody | anti-HA (mouse monoclonal) | Life Technologies | 26183 (RRID:AB_10978021) | 1:1000 |
| Antibody | Anti-HA high affinity, (rat monoclonal) | Roche | 11867423001 (RRID:AB_390918) | 1:1000 |
| Antibody | anti-Rab12 (rabbit polyclonal) | ProteinTech | 18843–1-AP (RRID:AB_10603469) | 1:1000 |
| Antibody | anti-Rab12 (sheep polyclonal) | MRC PPU Reagents and Services, University of Dundee | SA227 (AB_2921227) | 1 µg/ml |
| Antibody | anti-Rab12 phospho S106 (rabbit monoclonal) | Abcam | ab256487 (RRID:AB_2884880) | 1:1000 |
| Antibody | anti-PPM1H (sheep polyclonal) | MRC PPU Reagents and Services, University of Dundee | DA018 (RRID:AB_2923281) | 1:1000 |
| Antibody | anti-LC3A/B (rabbit polyclonal) | Cell Signaling Technology | 4108 (RRID:AB_2137703) | 1:1000 |
| Antibody | anti-GFP (chicken polyclonal) | Aves | GFP-1020 (RRID:AB_10000240) | 1:5000 |
| Antibody | anti-Arl13b (mouse monoclonal) | Neuromab | N295B/66 | 1:2000 |
| Antibody | Goat anti-Rabbit 800 (Goat polyclonal) | Licor | RRID: AB_621843 | 1:10000 |
| Antibody | Goat anti-Mouse 680 (Goat polyclonal) | Licor | RRID: AB_10956588 | 1:10000 |
| Antibody | Donkey anti-Rabbit 680 (Donkey polyclonal) | Licor | RRID: AB_10954442 | 1:10000 |
| Antibody | Donkey anti-Mouse 680 (Donkey polyclonal) | Licor | RRID: AB_10953628 | 1:10000 |
| Antibody | Donkey anti-Chicken 680 (Donkey polyclonal) | Licor | RRID: AB_10974977 | 1:10000 |

*Appendix 1 Continued on next page*

*Appendix 1 Continued*

| Reagent type (species) or resource | Designation | Source or reference | Identifiers | Additional information |
|---|---|---|---|---|
| Antibody | Rabbit anti-Sheep 800 (Rabbit polyclonal) | Invitrogen | RRID: AB_2556640 | 1:10000 |
| Antibody | Donkey anti-sheep 680 (Donkey polyclonal) | Life Technologies | RRID: AB_2535755 | 1:10000 |
| Antibody | Goat-anti chicken 680 (Goat polyclonal) | Life Technologies | RRID: AB_2762846 | 1:10000 |
| Antibody | Donkey anti-rabbit Alexa 647 H+L (Donkey polyclonal) | Life Technologies | RRID: AB_2536183 | 1:2000 |
| Antibody | Donkey anti-rabbit Alexa 568 H+L (Donkey polyclonal) | Life Technologies | RRID; AB_2534017 | 1:2000 |
| Antibody | Donkey anti-mouse Alexa 488 (Donkey polyclonal) | Life Technologies | RRID: AB_141607 | 1:2000 |
| Antibody | Donkey anti-mouse Alexa 555 (Donkey polyclonal) | Life Technologies | RRID: AB_2762848 | 1:2000 |
| Antibody | Donkey anti-mouse Alexa 647 (Donkey polyclonal) | Life Technologies | RRID: AB_2762830 | 1:2000 |
| Cell line (human) | HeLa | ATCC | CCL-2 RRID:CVCL_0030 | |
| Cell line (human) | HEK293T | ATCC | CRL-3216 RRID:CVCL_0063 | |
| Cell line (human) | HEK293 | ATCC | CRL-1573 (RRID: CVCL_0045) | |
| Cell line (mouse) | NIH-3T3-flpin | Life Technologies | R76107 (RRID:CVCL_U422) | |
| Cell line (human) | A549 | ATCC | ATCC-CCL-185 (RRID:CVCL_0023) | |
| Cell line (human) | hTERT-RPE | ATCC | ATCC-CRL-4000 (RRID:CVCL_4388) | |
| Cell line (human) | A549-PPM1H KO | MRC-PPU | In process | PMIID: 31663853 |
| Cell line (human) | A549-LRRK2 KO | MRC-PPU | In process | |
| Cell line (mouse) | MEF WT | MRC-PPU | Generated from RRID: MMRRC_049312-UCD | |
| Cell line (mouse) | MEF Rab12 KO | MRC-PPU | Generated from RRID: MMRRC_049312-UCD | |
| Strain, strain background (*E. coli*) | *E. coli* STBL3 | Thermo Fisher | C737303 | |
| Bacterial strain | Endura DUOs | Biosearch Technologies | 60242–1 | |
| Strain, strain background (*E. coli*) | *E. coli* Dh5a | Life Technologies | 18258012 | |
| Commercial Assay or Kit | 4–20% precast gels | Biorad | 4561096 | |
| Commercial Assay or Kit | MycoAlert detection kit | Lonza | LT07-318 | |
| Commercial Assay or Kit | RED-NHS 2nd Generation (Amine Reactive) Protein Labeling Kit | Nanotemper | MO-L011 | |
| Chemical compound, drug | Puromycin | Invivogen | Ant-pr-1 | Use at 1 µg/ml |

*Appendix 1 Continued on next page*

*Appendix 1 Continued*

| Reagent type (species) or resource | Designation | Source or reference | Identifiers | Additional information |
|---|---|---|---|---|
| Chemical compound, drug | Blasticidin | Invivogen | Ant-bl-1 | Use at 10 µg/ml |
| Chemical compound, drug | MLi-2 | MRC PPU Reagents and Services, University of Dundee | Cas No.: 1627091-47-7 | |
| Chemical compound, drug | L-Leucyl-L-Leucine methyl ester (hydrochloride) (LLOME) | Cayman Chemical | #16008 | |
| Chemical compound, drug | Nigericin | Invivogen | NC0813465 | 1–5 µM for 2–4 hrs |
| Chemical compound, drug | DMEM high glucose | Cytiva | SH30243.02 | |
| Chemical compound, drug | Penicillin/Streptomycin | Cytiva | SV30010 | |
| Chemical compound, drug | Fetal calf serum | Sigma | F0926 | |
| Chemical compound, drug | Glutamax | Thermo Scientific | 35050061 | |
| Chemical compound, drug | Gotaq 2 x | Promega | M7122 | |
| Chemical compound, drug | Titanium taq | Takara bio | NC9806143 | |
| Chemical compound, drug | Ex-taq | Takara bio | RR01CM | |
| Chemical compound, drug | NEB next 2 x | NEB | E7649AVIAL | |
| Chemical compound, drug | Proteinase K | Qiagen | 19133 | |
| Chemical compound, drug | RNaseH | ThermoFisher | 18021014 | |
| Commercial Assay or Kit | AL buffer | Qiagen | 19075 | |
| Commercial Assay or Kit | AW1 buffer | Qiagen | 19081 | |
| Commercial Assay or Kit | AW2 buffer | Qiagen | 19072 | |
| Commercial Assay or Kit | Econospin column | Epoch lifesciences | 1920-050/250 | |
| Commercial Assay or Kit | QuickExtract | Lucigen | QE09050 | |
| Commercial Assay or Kit | Ampure beads | Beckman | A63880 | |
| Recombinant DNA reagent | Lenti-guide puro | Addgene | RRID:Addgene_52963 | |
| Recombinant DNA reagent s | pMCB306 | Addgene | RRID:Addgene_89360 | |
| Recombinant DNA reagent | gRNA library (BRIE) | Addgene | RRID:Addgene_73633 | |
| Recombinant DNA reagent | Lenti-Cas9-blast | Addgene | RRID:Addgene_52962 | |
| Recombinant DNA reagent | pMCB306 GFP-Rab8A | Addgene | RRID:Addgene_198470 | PMID: 29125462 |

*Appendix 1 Continued on next page*

*Appendix 1 Continued*

| Reagent type (species) or resource | Designation | Source or reference | Identifiers | Additional information |
|---|---|---|---|---|
| Recombinant DNA reagent | pMCB306 GFP-Rab10 | Addgene | RRID:Addgene_130883 | |
| Recombinant DNA reagent | pMCB306 GFP-Rab12 | Addgene | RRID:Addgene_198471 | |
| Recombinant DNA reagent | pMCB306 GFP-Rab29 | Addgene | RRID:Addgene_198472 | PMID: 31624137 |
| Recombinant DNA reagent | pCMV5D HA-PPM1H | MRC PPU Reagents and Services, University of Dundee | DU62789 | |
| Recombinant DNA reagent | pCMV5D HA-PPM1H H153D | MRC PPU Reagents and Services, University of Dundee | DU62928 | |
| Recombinant DNA reagent | pCMV5D HA-PPM1H D288A | MRC PPU Reagents and Services, University of Dundee | DU62985 | |
| Recombinant DNA reagent | Lenti-guide-puro mRab12 | Addgene | RRID:Addgene_198475 RRID:Addgene_198476 | |
| Recombinant DNA reagent | Lenti-guide-puro mAtp6v1a | Addgene | RRID:Addgene_198477 RRID:Addgene_198478 | |
| Recombinant DNA reagent | Lenti-guide-puro mAtp5c | Addgene | RRID:Addgene_198479 RRID:Addgene_198480 | |
| Recombinant DNA reagent | Lenti-guide-puro mHgs | Addgene | RRID:Addgene_198481 RRID:Addgene_198482 | |
| Recombinant DNA reagent | Lenti-guide-puro mPHB2 | Addgene | RRID:Addgene_198483 RRID:Addgene_198484 | |
| Recombinant DNA reagent | Lenti-guide-puro mBltp1 (KIAA1109) | Addgene | RRID:Addgene_198489 RRID:Addgene_198490 | |
| Recombinant DNA reagent | Lenti-guide-puro mMyh9 | Addgene | RRID:Addgene_198491 RRID:Addgene_198492 | |
| Recombinant DNA reagent | Lenti-guide-puro mSptlc2 | Addgene | RRID:Addgene_198494 | |
| Recombinant DNA reagent | Lenti-guide-puro mYwhae | Addgene | RRID:Addgene_198497 RRID:Addgene_198498 | |
| Recombinant DNA reagent | Lenti-guide-puro mNudcd3 | Addgene | RRID:Addgene_198501 RRID:Addgene_198502 | |
| Recombinant DNA reagent | Lenti-guide-puro mCct8 | Addgene | RRID:Addgene_198503 RRID:Addgene_198504 | |
| Recombinant DNA reagent | Lenti-guide-puro mCsnk2b | Addgene | RRID:Addgene_198505 RRID:Addgene_198506 | |
| Recombinant DNA reagent | PSPAX2 | Addgene | RRID:Addgene_12260 | |
| Recombinant DNA reagent | VSV-G | Addgene | RRID:Addgene_12259 | |
| Recombinant DNA reagent | pCMV5 Flag-LRRK2 wild-type | MRC PPU Reagents and Services, University of Dundee | DU62804 | |
| Recombinant DNA reagent | pCMV5 Flag-LRRK2 R1441C | MRC PPU Reagents and Services, University of Dundee | DU13078 | |
| Recombinant DNA reagent | pCMV5 Flag-LRRK2 G2019S | MRC PPU Reagents and Services, University of Dundee | DU10129 | |
| Recombinant DNA reagent | pCMV5 Flag-LRRK2 K17/18 A R1441G | Addgene RRID:Addgene_186012 | 186012 | |
| Recombinant DNA reagent | pCMV5 Flag-LRRK2 D2017A | MRC PPU Reagents and Services, University of Dundee | DU10128 | |

*Appendix 1 Continued on next page*

*Appendix 1 Continued*

| Reagent type (species) or resource | Designation | Source or reference | Identifiers | Additional information |
|---|---|---|---|---|
| Recombinant DNA reagent | pCMV5 Flag-LRRK2 E240A | MRC PPU Reagents and Services, University of Dundee | DU72874 | |
| Recombinant DNA reagent | pCMV5 Flag-LRRK2 E240R | MRC PPU Reagents and Services, University of Dundee | DU72829 | |
| Recombinant DNA reagent | pCMV5 Flag-LRRK2 V241A | MRC PPU Reagents and Services, University of Dundee | DU72806 | |
| Recombinant DNA reagent | pCMV5 Flag-LRRK2 V241R | MRC PPU Reagents and Services, University of Dundee | DU72807 | |
| Recombinant DNA reagent | pCMV5 Flag-LRRK2 M243A | MRC PPU Reagents and Services, University of Dundee | DU72847 | |
| Recombinant DNA reagent | pCMV5 Flag-LRRK2 S244R | MRC PPU Reagents and Services, University of Dundee | DU72808 | |
| Recombinant DNA reagent | pCMV5 Flag-LRRK2 N246A | MRC PPU Reagents and Services, University of Dundee | DU72779 | |
| Recombinant DNA reagent | pCMV5 Flag-LRRK2 N246D | MRC PPU Reagents and Services, University of Dundee | DU72820 | |
| Recombinant DNA reagent | pCMV5 Flag-LRRK2 F283A | MRC PPU Reagents and Services, University of Dundee | DU72868 | |
| Recombinant DNA reagent | pCMV5 Flag-LRRK2 I285A | MRC PPU Reagents and Services, University of Dundee | DU72821 | |
| Recombinant DNA reagent | pCMV5 Flag-LRRK2 L286D | MRC PPU Reagents and Services, University of Dundee | DU72809 | |
| Recombinant DNA reagent | pCMV5 Flag-LRRK2 R399E | MRC PPU Reagents and Services, University of Dundee | DU72192 | |
| Recombinant DNA reagent | pCMV5 Flag-LRRK2 L403E | MRC PPU Reagents and Services, University of Dundee | DU72194 | |
| Recombinant DNA reagent | pCMV5 HA-empty | MRC PPU Reagents and Services, University of Dundee | DU49302 | |
| Recombinant DNA reagent | pCMV5 HA-Rab29 wild-type | MRC PPU Reagents and Services, University of Dundee | DU50222 | |
| Recombinant DNA reagent | pCMV5 HA-Rab12 wild-type | MRC PPU Reagents and Services, University of Dundee | DU48963 | |
| Recombinant DNA reagent | pCMV5 HA-Rab12 S106A | MRC PPU Reagents and Services, University of Dundee | DU48966 | |
| Recombinant DNA reagent | pCMV5 HA-Rab12 S106E | MRC PPU Reagents and Services, University of Dundee | DU48967 | |
| Recombinant DNA reagent | pQE-80L 2xHis Rab12 Q101L | Addgene in progress | | |
| Recombinant DNA reagent | pQE-80L 2xHis Armadillo E240R | Addgene in progress | | |
| Recombinant DNA reagent | pQE-80L 2xHis Armadillo K439E | Addgene in progress | | |
| Software, Algorithm | Jupyter notebook | Open source web application | RRID:SCR_018315 | |
| Software, Algorithm | Python | Programming language | RRID:SCR_008394 | |
| Commercial assay, kit | MiSeq v2 (300) | Illumina | MS-102–2002 | |
| Software, Algorithm | CellProfiler | PMID: 29969450 | RRID:SCR_007358 | |
| Software, Algorithm | MAGeCK | PMID: 25476604 | | |
| Software, Algorithm | Chimera X | PMID: 32881101 | RRID:SCR_015872 | |

*Appendix 1 Continued on next page*

*Appendix 1 Continued*

| Reagent type (species) or resource | Designation | Source or reference | Identifiers | Additional information |
|---|---|---|---|---|
| Software, Algorithm | Prism | Prism 9 version 9.3.1 (350) | RRID:SCR_002798 | |
| Software, Algorithm | R CRAN R package ggridges_0.5.3 | version 4.2.0 (2022-04-22) | RRID:SCR_001905 | |
| Software, Algorithm | Dplyr | Version 1.0.9 | RRID:SCR_016708 | |
| Software, Algorithm | ggplot | Version 3.3.6 | RRID:SCR_014601 | |
| Software, Algorithm | ImageJ | Version 1.53 v | RRID:SCR_003070 | |
| Software, Algorithm | Metamorph | | RRID:SCR_002368 | |
| Software, Algorithm | Fiji | Version 2017 May 30 | RRID:SCR_002285 | |
| Software, Algorithm | Adobe Illustrator | Version 27.2 | RRID:SCR_010279 | |
| Software, Algorithm | ImageStudioLite | Version 5.2.5 | RRID:SCR_013715 | |
| Software, Algorithm | NanoTemper NTAAffinityAnalysis | MO.Affinity Analysis v2.2.5 | | |

