## [Editor Report]

LRRK2 is a multi-domain kinase and is known to phosphorylate a subset of Rab proteins involved in intracellular trafficking, and Parkinson's disease-linked mutations increase this phosphorylation. How LRRK2 becomes activated is a major question in the field. This highly interesting work adds a new layer to our mechanistic understanding of this complex protein, revealing that binding of Rab12 to LRRK2 stimulates its ability to phosphorylate Rab10, a conclusion that is supported by extensive and robust evidence from a wide array of approaches.

---

## [Decision Letter]

**Decision letter after peer review:**

Thank you for submitting your article "Genome-wide screen reveals Rab12 GTPase as a critical activator of pathogenic LRRK2 kinase" for consideration by *eLife*. Your article has been reviewed by 3 peer reviewers, and the evaluation has been overseen by a Reviewing Editor and Vivek Malhotra as the Senior Editor. The following individual involved in the review of your submission has agreed to reveal their identity: Andres E Leschziner (Reviewer #3).

Essential revisions:

Overall the reviewers think that the paper is a strong contribution to the field. There are two comments from the reviewers that we feel would enhance the paper if you are able to do the experiments.

First, would it be possible to perform binding assays in cells with full-length LRRK2 to show that its residues E240 and S244 actually mediate binding to Rab12? Presumably, this would be a relatively straightforward IP-western.

Second, besides disrupting mutations, you also found one mutation (F283A) enhancing the cellular effect of Rab12 overexpression demonstrated by increased pRab10 levels. For a better evaluation of the presented computational model of the Rab12:LRRK2 complex, it would be interesting, if you could examine the binding of the F283A mutant as well.

The reviewers noted several other issues, which we feel could be largely handled by changes to the text, but of course, if you have data for any of these points, it could be added.

*Reviewer #2 (Recommendations for the authors):*

1) Besides disrupting mutations, the authors also found one mutation enhancing the cellular effect of Rab12 overexpression demonstrated by increased pRab10 levels. For a better evaluation of the presented computational model of the Rab12:LRRK2 complex, it would be interesting, if the authors could study the binding affinity of that mutant (F283A), as well.

2) Given that the presented structure of the Rab12:LRRK2 complex is based on computational modelling, alternative docking poses should be presented, e.g. as overlap, to allow the reader to estimate the modelling precision.

3) Sharing a PDB file of the final model(s) might be helpful for the community.

*Reviewer #3 (Recommendations for the authors):*

General

– The manuscript would be easier to follow by a wider audience if the authors were to add a figure (as Figure 1A) showing LRRK2, Rab12, and Rab10. Either in their primary structures or as cartoons of their 3D structures, highlighting known interaction sites, mutations, and PTMs.

– Can the authors comment on Rab8 in the context of this manuscript? Do the levels of Rab12 also affect pRab8A as well or is this a Rab10-specific effect?

– Did the authors analyze the phosphorylation profile of Rab12? Does it have to be phosphorylated to activate LRRK2? Is it phosphorylated by LRRK2? Data addressing these questions would be a nice addition to this manuscript.

Title

– I find the title somewhat misleading. I worry that people could interpret it as meaning that Rab12 "only" activates LRRK2 carrying Parkinson's disease-linked mutations and not the wild-type protein.

Abstract

– "(…) we showed previously that phosphoRabs play an important role in LRRK2 membrane recruitment and activation."

The authors should change this to 'certain phopshoRabs' or 'a subset of phosphoRabs' to prevent confusion as this statement implies that all phosphorylated Rabs are important in LRRK2 membrane recruitment and activation. This would not be a problem with people in the field but may confuse a wider audience.

– "AlphaFold modeling revealed a novel Rab12 binding site in the LRRK2 Armadillo domain and we show that residues predicted to be essential for Rab12 interaction at this site influence overall phosphoRab levels in a manner distinct from Rab29 activation of LRRK2."

Same as above, the authors should be more specific about what Rabs they are referring to.

Introduction

– I found that paragraph 3 breaks the flow of the Introduction somewhat in its present form. The paragraph does introduce important information for the paragraph that follows, so I would suggest expanding it a bit to make it more accessible to a wider audience. The schematic figure I suggested above, which should indicate where sites #1 and #2 are located in LRRK2, would help readers follow the information here.

Results section

– Both "NIH-3T3" and "3T3" are used to refer to cell one. It would help to choose one for consistency.

– "Fixed cells are stained with an antibody that specifically and sensitively detects phosphoRab10 and then sorted by flow cytometry to separate cells based on phosphoRab10 content."

Presumably, this is specific for pT73? If that's the case, the authors should specify that.

– "NUDCD3 stabilizes the dynein intermediate chain and is likely important for concentrating phosphoRab10 at the mother centriole."

Please add a reference.

– "AlphaFold (Jumper et al., 2021) in conjunction with Colabfold in ChimeraX (Mirdita et al., 2022; Pettersen et al., 2004) revealed a third Rab binding site (Site #3) when the full-length Armadillo domain was modeled together with Rab12 (Figure 6A; see Figure 9 below)."

Please indicate the boundaries of the Armadillo domain used in this study.

---

## [Author Response]

Essential revisions:Overall the reviewers think that the paper is a strong contribution to the field. There are two comments from the reviewers that we feel would enhance the paper if you are able to do the experiments.First, would it be possible to perform binding assays in cells with full-length LRRK2 to show that its residues E240 and S244 actually mediate binding to Rab12? Presumably, this would be a relatively straightforward IP-western.Second, besides disrupting mutations, you also found one mutation (F283A) enhancing the cellular effect of Rab12 overexpression demonstrated by increased pRab10 levels. For a better evaluation of the presented computational model of the Rab12:LRRK2 complex, it would be interesting, if you could examine the binding of the F283A mutant as well.The reviewers noted several other issues, which we feel could be largely handled by changes to the text, but of course, if you have data for any of these points, it could be added.

We thank the reviewers for their positive assessment and have carried out all the requested experiments as described below.

Reviewer #2 (Recommendations for the authors):1) Besides disrupting mutations, the authors also found one mutation enhancing the cellular effect of Rab12 overexpression demonstrated by increased pRab10 levels. For a better evaluation of the presented computational model of the Rab12:LRRK2 complex, it would be interesting, if the authors could study the binding affinity of that mutant (F283A), as well.

As requested, we carried out microscale thermophoresis of the ARM domain with that mutation. The mutant binds as well to Rab12 as the wild type protein (Figure 8D).

2) Given that the presented structure of the Rab12:LRRK2 complex is based on computational modelling, alternative docking poses should be presented, e.g. as overlap, to allow the reader to estimate the modelling precision.

We include an overlay of the top 5 predicted structures and a metric of confidence which is very high (New Figure 6—Figure Supp. 1).

3) Sharing a PDB file of the final model(s) might be helpful for the community.

A PDB file is now included as requested and a video showing overlay of Rab12 bound to ARM domain upon the full length LRRK2 structure.

Reviewer #3 (Recommendations for the authors):General– The manuscript would be easier to follow by a wider audience if the authors were to add a figure (as Figure 1A) showing LRRK2, Rab12, and Rab10. Either in their primary structures or as cartoons of their 3D structures, highlighting known interaction sites, mutations, and PTMs.

Shown now in new Figure 6.

– Can the authors comment on Rab8 in the context of this manuscript? Do the levels of Rab12 also affect pRab8A as well or is this a Rab10-specific effect?

We don’t have an antibody that allows us to monitor uniquely pRab8 as the antibody cross reacts with other phosphoRabs. We have added this to the text.

– Did the authors analyze the phosphorylation profile of Rab12? Does it have to be phosphorylated to activate LRRK2? Is it phosphorylated by LRRK2? Data addressing these questions would be a nice addition to this manuscript.

Phosphomimetic Rab mutants are poor analogs but as requested we show that activation does not require Rab12 phosphorylation, nor does it require the phosphoRab binding Site #2 (New Figure 3G,H). Rab12 is a LRRK2 substrate (Steger et al. 2016), pRab12 levels go down with MLi-2 addition.

Title– I find the title somewhat misleading. I worry that people could interpret it as meaning that Rab12 "only" activates LRRK2 carrying Parkinson's disease-linked mutations and not the wild-type protein.

Good point—we have modified the title as requested.

Abstract– "(…) we showed previously that phosphoRabs play an important role in LRRK2 membrane recruitment and activation."The authors should change this to 'certain phopshoRabs' or 'a subset of phosphoRabs' to prevent confusion as this statement implies that all phosphorylated Rabs are important in LRRK2 membrane recruitment and activation. This would not be a problem with people in the field but may confuse a wider audience.– "AlphaFold modeling revealed a novel Rab12 binding site in the LRRK2 Armadillo domain and we show that residues predicted to be essential for Rab12 interaction at this site influence overall phosphoRab levels in a manner distinct from Rab29 activation of LRRK2."Same as above, the authors should be more specific about what Rabs they are referring to.

Done

Introduction– I found that paragraph 3 breaks the flow of the Introduction somewhat in its present form.

Agreed! Thanks!

The paragraph does introduce important information for the paragraph that follows, so I would suggest expanding it a bit to make it more accessible to a wider audience. The schematic figure I suggested above, which should indicate where sites #1 and #2 are located in LRRK2, would help readers follow the information here.

We have made the text more general and moved forward the model from Figure 9 to Figure 6 to make the details clearer for a general reader.

Results section– Both "NIH-3T3" and "3T3" are used to refer to cell one. It would help to choose one for consistency.

Thanks

– "Fixed cells are stained with an antibody that specifically and sensitively detects phosphoRab10 and then sorted by flow cytometry to separate cells based on phosphoRab10 content."Presumably, this is specific for pT73? If that's the case, the authors should specify that.

Thanks

– "NUDCD3 stabilizes the dynein intermediate chain and is likely important for concentrating phosphoRab10 at the mother centriole."Please add a reference.

Done

– "AlphaFold (Jumper et al., 2021) in conjunction with Colabfold in ChimeraX (Mirdita et al., 2022; Pettersen et al., 2004) revealed a third Rab binding site (Site #3) when the full-length Armadillo domain was modeled together with Rab12 (Figure 6A; see Figure 9 below)."Please indicate the boundaries of the Armadillo domain used in this study.

Done. Thanks!